# Benefit of deep learning with non-convex noisy gradient descent: Provable excess risk bound and superiority to kernel methods

**Taiji Suzuki**
Graduate School of Information Science and Technology, The University of Tokyo, Japan
Center for Advanced Intelligence Project, RIKEN, Japan
E-mail: `taiji@mist.i.u-tokyo.ac.jp`

**Shunta Akiyama**
Graduate School of Information Science and Technology, The University of Tokyo, Japan
E-mail: `akiyama@mist.i.u-tokyo.ac.jp`

## Abstract

Establishing a theoretical analysis that explains why deep learning can outperform shallow learning such as kernel methods is one of the biggest issues in the deep learning literature. Towards answering this question, we evaluate excess risk of a deep learning estimator trained by a noisy gradient descent with ridge regularization on a mildly overparameterized neural network, and discuss its superiority to a class of linear estimators that includes neural tangent kernel approach, random feature model, other kernel methods, $k$-NN estimator and so on. We consider a teacher-student regression model, and eventually show that *any* linear estimator can be outperformed by deep learning in a sense of the minimax optimal rate especially for a high dimension setting. The obtained excess bounds are so-called fast learning rate which is faster than $O(1/\sqrt{n})$ that is obtained by usual Rademacher complexity analysis. This discrepancy is induced by the non-convex geometry of the model and the noisy gradient descent used for neural network training provably reaches a near global optimal solution even though the loss landscape is highly non-convex. Although the noisy gradient descent does not employ any explicit or implicit sparsity inducing regularization, it shows a preferable generalization performance that dominates linear estimators.

## 1 Introduction

In the deep learning theory literature, clarifying the mechanism by which deep learning can outperform shallow approaches has been gathering most attention for a long time. In particular, it is quite important to show that a tractable algorithm for deep learning can provably achieve a better generalization performance than shallow methods. Towards that goal, we study the rate of convergence of *excess risk* of both deep and shallow methods in a setting of a nonparametric regression problem. One of the difficulties to show generalization ability of deep learning with certain optimization methods is that the solution is likely to be stacked in a bad local minimum, which prevents us to show its preferable performances. Recent studies tackled this problem by considering optimization on overparameterized networks as in neural tangent kernel (NTK) (Jacot et al., 2018; Du et al., 2019a) and mean field analysis (Nitanda & Suzuki, 2017; Chizat & Bach, 2018; Rotskoff & Vanden-Eijnden, 2018; 2019; Mei et al., 2018; 2019), or analyzing the noisy gradient descent such as stochastic gradient Langevin dynamics (SGLD) (Welling & Teh, 2011; Raginsky et al., 2017; Erdogdu et al., 2018).

The NTK analysis deals with a relatively large scale initialization so that the model is well approximated by the tangent space at the initial solution, and eventually, all analyses can be reduced to those of kernel methods (Jacot et al., 2018; Du et al., 2019b; Allen-Zhu et al., 2019; Du et al., 2019a; Arora et al., 2019; Cao & Gu, 2019; Zou et al., 2020). Although this regime is useful to show its global

convergence, the obtained estimator looses large advantage of deep learning approaches because the estimation ability is reduced to the corresponding kernel methods. To overcome this issue, there are several "beyond-kernel" type analyses. For example, Allen-Zhu & Li (2019; 2020) showed benefit of depth by analyzing ResNet type networks. Li et al. (2020) showed global optimality of gradient descent by reducing the optimization problem to a tensor decomposition problem for a specific regression problem, and showed the "ideal" estimator on a linear model has worse dependency on the input dimensionality. Bai & Lee (2020) considered a second order Taylor expansion and showed that the sample complexity of deep approaches has better dependency on the input dimensionality than kernel methods. Chen et al. (2020) also derived a similar conclusion by considering a hierarchical representation. The analyses mentioned above actually show some superiority of deep learning, but all of these bounds are essentially $\Omega(1/\sqrt{n})$ where $n$ is the sample size, which is not optimal for regression problems with squared loss (Caponnetto & de Vito, 2007). The reason why only such a sub-optimal rate is considered is that the target of their analyses is mostly the Rademacher complexity of the set in which estimators exist for bounding the generalization gap. However, to derive a tight *excess risk* bound instead of the generalization gap, we need to evaluate so called *local Rademacher complexity* (Mendelson, 2002; Bartlett et al., 2005; Koltchinskii, 2006) (see Eq. (2) for the definition of excess risk). Moreover, some of the existing analyses should change the target function class as the sample size $n$ increases, for example, the input dimensionality is increased against the sample size, which makes it difficult to see how the rate of convergence is affected by the choice of estimators.

Another promising approach is the mean field analysis. There are also some work that showed superiority of deep learning against kernel methods. Ghorbani et al. (2019) showed that, when the dimensionality $d$ of input is polynomially increasing with respect to $n$, the kernel methods is outperformed by neural network approaches. Although the situation of increasing $d$ explains well the modern high dimensional situations, this setting blurs the rate of convergence. Actually, we can show the superiority of deep learning even in a *fixed dimension* setting.

There are several studies about approximation abilities of deep and shallow models. Ghorbani et al. (2020) showed adaptivity of kernel methods to the intrinsic dimensionality in terms of approximation error and discuss difference between deep and kernel methods. Yehudai & Shamir (2019) showed that the random feature method requires exponentially large number of nodes against the input dimension to obtain a good approximation for a single neuron target function. These are only for approximation errors and estimation errors are not compared.

Recently, the superiority of deep learning against kernel methods has been discussed also in the nonparametric statistics literature where the minimax optimality of deep learning in terms of excess risk is shown. Especially, it is shown that deep learning achieves better rate of convergence than *linear estimators* in several settings (Schmidt-Hieber, 2020; Suzuki, 2019; Imaizumi & Fukumizu, 2019; Suzuki & Nitanda, 2019; Hayakawa & Suzuki, 2020). Here, the linear estimators are a general class of estimators that includes kernel ridge regression, $k$-NN regression and Nadaraya-Watson estimator. Although these analyses give clear statistical characterization on estimation ability of deep learning, they are not compatible with tractable optimization algorithms.

In this paper, we give a theoretical analysis that unifies these analyses and shows the superiority of a deep learning method trained by a tractable noisy gradient descent algorithm. We evaluate the excess risks of the deep learning approach and linear estimators in a nonparametric regression setting, and show that the minimax optimal convergence rate of the linear estimators can be dominated by the noisy gradient descent on neural networks. In our analysis, the model is fixed and no explicit sparse regularization is employed. Our contributions can be summarized as follows:

- A refined analysis of excess risks for a fixed model with a fixed input dimension is given to compare deep and shallow estimators. Although several studies pointed out the curse of dimensionality is a key factor that separates shallow and deep approaches, we point out that such a separation appears in a rather low dimensional setting, and more importantly, the *non-convexity* of the model essentially makes the two regimes different.

- A lower bound of the excess risk which is valid for *any* linear estimator is derived. The analysis is considerably general because the class of linear estimators includes kernel ridge regression with any kernel and thus it also includes estimators in the NTK regime.

- All derived convergence rate is a fast learning rate that is faster than $O(1/\sqrt{n})$. We show that simple noisy gradient descent on a sufficiently wide two-layer neural network achieves a fast

learning rate by using a fact that the solution converges to a Bayes estimator with a Gaussian process prior, and the derived convergence rate can be faster than that of linear estimators. This is much different from such existing work that compared only coefficients with the same rate of convergence with respect to the sample size $n$.

**Other related work** Bach (2017) analyzed the model capacity of neural networks and its corresponding reproducing kernel Hilbert space (RKHS), and showed that the RKHS is much larger than the neural network model. However, separation of the estimation abilities between shallow and deep is not proven. Moreover, the analyzed algorithm is basically the Frank-Wolfe type method which is not typically used in practical deep learning. The same technique is also employed by Barron (1993). The Frank-Wolfe algorithm is a kind of sparsity inducing algorithm that is effective for estimating a function in a model with an $L_1$-norm constraint. It has been shown that explicit or implicit sparse regularization such as $L_1$-regularization is beneficial to obtain better performances of deep learning under certain situations (Chizat & Bach, 2020; Chizat, 2019; Gunasekar et al., 2018; Woodworth et al., 2020; Klusowski & Barron, 2016). For example, E et al. (2019b;a) showed that the approximation error of a linear model suffers from the curse of dimensionality in a setting where the target function is in the Barron class (Barron, 1993), and showed an $L_1$-type regularization avoids the curse of dimensionality. However, our analysis goes in a different direction where a sparse regularization is not required.

## 2 PROBLEM SETTING AND MODEL

In this section, we give the problem setting and notations that will be used in the theoretical analysis. We consider the standard nonparametric regression problem where data are generated from the following model for an unknown true function $f^{\mathrm{o}} : \mathbb{R}^d \to \mathbb{R}$:

$$y_i = f^{\mathrm{o}}(x_i) + \epsilon_i \quad (i = 1, \ldots, n), \tag{1}$$

where $x_i$ is independently identically distributed from $P_X$ whose support is included in $\Omega = [0, 1]^d$, and $\epsilon_i$ is an observation noise that is independent of $x_i$ and satisfies $\mathrm{E}[\epsilon_i] = 0$ and $\epsilon_i \in [-U, U]$ almost surely. The $n$ i.i.d. observations are denoted by $D_n = (x_i, y_i)_{i=1}^n$. We want to estimate the true function $f^{\mathrm{o}}$ through the training data $D_n$. To achieve this purpose, we employ the squared loss $\ell(y, f(x)) = (y - f(x))^2$ and accordingly we define the expected and empirical risks as $\mathcal{L}(f) := \mathrm{E}_{Y,X}[\ell(Y, f(X))]$ and $\widehat{\mathcal{L}}(f) := \frac{1}{n} \sum_{i=1}^n \ell(y_i, f(x_i))$ respectively. Throughout this paper, we are interested in the *excess (expected) risk* of an estimator $\widehat{f}$ defined by

$$\text{(Excess risk)} \qquad \mathcal{L}(\widehat{f}) - \inf_{f:\text{measurable}} \mathcal{L}(f). \tag{2}$$

Since the loss function $\ell$ is the squared loss, the infimum of $\inf_{f:\text{measurable}} \mathcal{L}(f)$ is achieved by $f^{\mathrm{o}}$: $\inf_{f:\text{measurable}} \mathcal{L}(f) = \mathcal{L}(f^{\mathrm{o}})$. The population $L_2(P_X)$-norm is denoted by $\|f\|_{L_2(P_X)} := \sqrt{\mathrm{E}_{X \sim P_X}[f(X)^2]}$ and the sup-norm on the support of $P_X$ is denoted by $\|f\|_\infty := \sup_{x \in \mathrm{supp}(P_X)} |f(x)|$. We can easily check that for an estimator $\widehat{f}$, the $L_2$-distance $\|\widehat{f} - f^{\mathrm{o}}\|_{L_2(P_X)}^2$ between the estimator $\widehat{f}$ and the true function $f^{\mathrm{o}}$ is identical to the excess risk: $\mathcal{L}(\widehat{f}) - \mathcal{L}(f^{\mathrm{o}}) = \|\widehat{f} - f^{\mathrm{o}}\|_{L_2(P_X)}^2$. Note that the excess risk is different from the generalization gap $\mathcal{L}(\widehat{f}) - \widehat{\mathcal{L}}(\widehat{f})$. Indeed, the generalization gap typically converges with the rate of $O(1/\sqrt{n})$ which is optimal in a typical setting (Mohri et al., 2012). On the other hand, the excess risk can be faster than $O(1/\sqrt{n})$, which is known as a *fast learning rate* (Mendelson, 2002; Bartlett et al., 2005; Koltchinskii, 2006; Giné & Koltchinskii, 2006).

### 2.1 MODEL OF TRUE FUNCTIONS

To analyze the excess risk, we need to specify a function class (in other words, model) in which the true function $f^{\mathrm{o}}$ is included. In this paper, we only consider a two layer neural network model, whereas the techniques adapted in this paper can be directly extended to deeper neural network models. We consider a teacher-student setting, that is, the true function $f^{\mathrm{o}}$ can be represented by a neural network defined as follows. For $w \in \mathbb{R}$, let $\bar{w}$ be a "clipping" of $w$ defined as $\bar{w} :=$

$R \times \tanh(w/R)$ where $R \geq 1$ is a fixed constant, and let $[x; 1] := [x^\top, 1]^\top$ for $x \in \mathbb{R}^d$. Then, the teacher network is given by

$$f_W(x) = \sum_{m=1}^\infty a_m \bar{w}_{2,m} \sigma_m(w_{1,m}^\top [x; 1]),$$

where $w_{1,m} \in \mathbb{R}^{d+1}$ and $w_{2,m} \in \mathbb{R}$ ($m \in \mathbb{N}$) are the trainable parameters (where $W = (w_{1,m}, w_{2,m})_{m=1}^\infty$), $a_m \in \mathbb{R}$ ($m \in \mathbb{N}$) is a fixed scaling parameter, and $\sigma_m : \mathbb{R} \to \mathbb{R}$ is an activation function for the $m$-th node. The reason why we applied the clipping operation to the parameter of the second layer is just for a technical reason to ensure convergence of Langevin dynamics. The dynamics is bounded in high probability in practical situations and the boundedness condition would be removed if further theoretical development of infinite dimensional Langevin dynamics would be achieved.

Let $\mathcal{H}$ be a set of parameters $W$ such that its squared norm is bounded: $\mathcal{H} := \{W = (w_{1,m}, w_{2,m})_{m=1}^\infty \mid \sum_{m=1}^\infty (\|w_{1,m}\|^2 + w_{2,m}^2) < \infty\}$. Define $\|W\|_{\mathcal{H}} := [\sum_{m=1}^\infty (\|w_{1,m}\|^2 + w_{2,m}^2)]^{1/2}$ for $W \in \mathcal{H}$. Let $(\mu_m)_{m=1}^\infty$ be a regularization parameter such that $\mu_m \searrow 0$. Accordingly we define $\mathcal{H}_\gamma := \{W \in \mathcal{H} \mid \|W\|_{\mathcal{H}_\gamma} < \infty\}$ where $\|W\|_{\mathcal{H}_\gamma} := [\sum_{m=1}^\infty \mu_m^{-\gamma} (\|w_{1,m}\|^2 + w_{2,m}^2)]^{1/2}$ for a given $0 < \gamma$. Throughout this paper, we analyze an estimation problem in which the true function is included in the following model:

$$\mathcal{F}_\gamma = \{f_W \mid W \in \mathcal{H}_\gamma, \ \|W\|_{\mathcal{H}_\gamma} \leq 1\}.$$

This is basically two layer neural network with infinite width. As assumed later, $a_m$ is assumed to decrease as $m \to \infty$. Its decreasing rate controls the capacity of the model. If the first layer parameters $(w_{1,m})_m$ are fixed, this model can be regarded as a variant of the unit ball of some reproducing kernel Hilbert space (RKHS) with basis functions $a_m \sigma_m(w_{1,m}^\top [x; 1])$. However, since the first layer $(w_{1,m})$ is also trainable, there appears significant difference between deep and kernel approaches. The Barron class (Barron, 1993; E et al., 2019b) is relevant to this function class. Indeed, it is defined as the convex hull of $w_2 \sigma(w_1^\top [x; 1])$ with norm constraints on $(w_1, w_2)$ where $\sigma$ is an activation function. On the other hand, we will put an explicit decay rate on $a_m$ and the parameter $W$ has an $L_2$-norm constraint, which makes the model $\mathcal{F}_\gamma$ smaller than the Barron class.

## 3 ESTIMATORS

We consider two classes of estimators and discuss their differences: linear estimators and deep learning estimator with noisy gradient descent (NGD).

**Linear estimator** A class of linear estimators, which we consider as a representative of "shallow" learning approach, consists of all estimators that have the following form:

$$\hat{f}(x) = \sum_{i=1}^n y_i \varphi_i(x_1, \ldots, x_n, x).$$

Here, $(\varphi_i)_{i=1}^n$ can be any measurable function (and $L_2(P_X)$-integrable so that the excess risk can be defined). Thus, they could be selected as the "optimal" one so that the corresponding linear estimator minimizes the worst case excess risk. Even if we chose such an optimal one, the worst case excess risk should be lower bounded by our lower bound given in Theorem 1. It should be noted that the linear estimator does not necessarily imply "linear model." The most relevant linear estimator in the machine learning literature is the kernel ridge regression: $\hat{f}(x) = Y^\top (K_X + \lambda I)^{-1} \mathbf{k}(x)$ where $K_X = (k(x_i, x_j))_{i,j=1}^n \in \mathbb{R}^{n \times n}$, $\mathbf{k}(x) = [k(x, x_1), \ldots, k(x, x_n)]^\top \in \mathbb{R}^n$ and $Y = [y_1, \ldots, y_n]^\top \in \mathbb{R}^n$ for a kernel function $k : \mathbb{R}^d \times \mathbb{R}^d \to \mathbb{R}$. Therefore, the ridge regression estimator in the NTK regime or the random feature model is also included in the class of linear estimators. The solution obtained in the early stopping criteria instead of regularization in the NTK regime under the squared loss is also included in the linear estimators. Other examples include the $k$-NN estimator and the Nadaraya-Watson estimator. All of them do not train the basis function in a nonlinear way, which makes difference from the deep learning approach. In the nonparametric statistics literature, linear estimators have been studied for estimating a wavelet series model. Donoho et al. (1990; 1996) have shown that a wavelet shrinkage estimator can outperform any linear estimator by showing suboptimality of linear estimators. Suzuki (2019) utilized such an argument to show superiority of deep learning but did not present any tractable optimization algorithm.

**Noisy Gradient Descent with regularization**  As for the neural network approach, we consider a noisy gradient descent algorithm. Basically, we minimize the following regularized empirical risk:

$$\widehat{\mathcal{L}}(f_W) + \tfrac{\lambda}{2}\|W\|^2_{\mathcal{H}_1}.$$

Here, we employ $\mathcal{H}_1$-norm as the regularizer. We note that the constant $\gamma$ controls the relative complexity of the true function $f^\circ$ compared to the typical solution obtained under the regularization. Here, we define a linear operator $A$ as $\lambda\|W\|_{\mathcal{H}_1} = W^\top A W$, that is, $AW = (\lambda\mu_m^{-1}w_{1,m}, \lambda\mu_m^{-1}w_{2,m})_{m=1}^\infty$. The regularized empirical risk can be minimized by noisy gradient descent as $W_{k+1} = W_k - \eta\nabla(\widehat{\mathcal{L}}(f_{W_k}) + \tfrac{\lambda}{2}\|W_k\|^2_{\mathcal{H}_1}) + \sqrt{\tfrac{2\eta}{\beta}}\xi_k$, where $\eta > 0$ is a step size and $\xi_k = (\xi_{k,(1,m)}, \xi_{k,(2,m)})_{m=1}^\infty$ is an infinite-dimensional Gaussian noise, i.e., $\xi_{k,(1,m)}$ and $\xi_{k,(2,m)}$ are independently identically distributed from the standard normal distribution (Da Prato & Zabczyk, 1996). Here, $\nabla\widehat{\mathcal{L}}(f_W) = \frac{1}{n}\sum_{i=1}^n 2(f_W(x_i) - y_i)(\bar{w}_{2,m}a_m[x_i;1]\sigma_m'(w_{1,m}^\top[x_i;1]), a_m\tanh'(w_{2,m}/R)\sigma_m(w_{1,m}^\top[x_i;1]))_{m=1}^\infty$. However, since $\nabla\|W_{k-1}\|^2_{\mathcal{H}_1}$ is unbounded which makes it difficult to show convergence, we employ the *semi-implicit Euler scheme* defined by

$$W_{k+1} = W_k - \eta\nabla\widehat{\mathcal{L}}(f_{W_k}) - \eta A W_{k+1} + \sqrt{\tfrac{2\eta}{\beta}}\xi_k \Leftrightarrow W_{k+1} = S_\eta\left(W_k - \eta\nabla\widehat{\mathcal{L}}(f_{W_k}) + \sqrt{\tfrac{2\eta}{\beta}}\xi_k\right), \quad (3)$$

where $S_\eta := (\mathrm{I}+\eta A)^{-1}$. It is easy to check that this is equivalent to the following update rule: $W_k = W_{k-1} - \eta\left(\nabla\widehat{\mathcal{L}}(f_{W_{k-1}}) + S_\eta A W_{k-1} + \sqrt{\tfrac{2\eta}{\beta}}\xi_{k-1}\right)$. Therefore, the implicit Euler scheme can be seen as a naive noisy gradient descent for minimizing the empirical risk with a slightly modified ridge regularization. This can be interpreted as a discrete time approximation of the following *infinite dimensional Langevin dynamics*:

$$\mathrm{d}W_t = -\nabla(\widehat{\mathcal{L}}(f_{W_t}) + \tfrac{\lambda}{2}\|W_t\|^2_{\mathcal{H}_1})\mathrm{d}t + \sqrt{2/\beta}\mathrm{d}\xi_t, \quad (4)$$

where $(\xi_t)_{t\geq 0}$ is the so-called cylindrical Brownian motion (see Da Prato & Zabczyk (1996) for the details). Its application and analysis for machine learning problems with non-convex objectives have been recently studied by, for example, Muzellec et al. (2020); Suzuki (2020).

The above mentioned algorithm is executed on an infinite dimensional parameter space. In practice, we should deal with a finite width network. To do so, we approximate the solution by a finite dimensional one: $W^{(M)} = (w_{1,m}, w_{2,m})_{m=1}^M$ where $M$ corresponds to the width of the network. We identify $W^{(M)}$ to the "zero-padded" infinite dimensional one, $W = (w_{1,m}, w_{2,m})_{m=1}^\infty$ with $w_{1,m} = 0$ and $w_{2,m} = 0$ for all $m > M$. Accordingly, we use the same notation $f_{W^{(M)}}$ to indicate $f_W$ with zero padded vector $W$. Then, the finite dimensional version of the update rule is given by $W_{k+1}^{(M)} = S_\eta^{(M)}\left(W_k^{(M)} - \eta\nabla\widehat{\mathcal{L}}(f_{W_k^{(M)}}) + \sqrt{\tfrac{2\eta}{\beta}}\xi_k^{(M)}\right)$, where $\xi_k^{(M)}$ is the Gaussian noise vector obtained by projecting $\xi_k$ to the first $M$ components and $S_\eta^{(M)}$ is also obtained in a similar way.

## 4  CONVERGENCE RATE OF ESTIMATORS

In this section, we present the excess risk bounds for linear estimators and the deep learning estimator. As for the linear estimators, we give its lower bound while we give an upper bound for the deep learning approach. To obtain the result, we setup some assumptions on the model.

**Assumption 1.**

(i) *There exists a constant $c_\mu$ such that $\mu_m \leq c_\mu m^{-2}$ $(m \in \mathbb{N})$.*

(ii) *There exists $\alpha_1 > 1/2$ such that $a_m \leq \mu_m^{\alpha_1}$ $(m \in \mathbb{N})$.*

(iii) *The activation functions $(\sigma_m)_m$ is bounded as $\|\sigma_m\|_\infty \leq 1$. Moreover, they are three times differentiable and their derivatives upto third order differentiation are uniformly bounded: $\exists C_\sigma$ such that $\|\sigma_m\|_{1,3} := \max\{\|\sigma_m'\|_\infty, \|\sigma_m''\|_\infty, \|\sigma_m'''\|_\infty\} \leq C_\sigma$ $(\forall m \in \mathbb{N})$.*

The first assumption (i) controls the strength of the regularization, and combined with the second assumption (ii) and definition of the model $\mathcal{F}_\gamma$, complexity of the model is controlled. If $\alpha_1$ and $\gamma$ are large, the model is less complicated. Indeed, the convergence rate of the excess risk becomes

faster if these parameters are large as seen later. The decay rate $\mu_m \leq c_\mu m^{-2}$ can be generalized as $m^{-p}$ with $p > 1$ but we employ this setting just for a technical simplicity for ensuring convergence of the Langevin dynamics. The third assumption (iii) can be satisfied by several activation functions such as the sigmoid function and the hyperbolic tangent. The assumption $\|\sigma_m\|_\infty \leq 1$ could be replaced by another one like $\|\sigma_m\|_\infty \leq C$, but we fix this scaling for simple presentation.

## 4.1 MINIMAX LOWER BOUND FOR LINEAR ESTIMATORS

Here, we analyze a lower bound of excess risk of linear estimators, and eventually we show that *any* linear estimator suffers from curse of dimensionality. To rigorously show that, we consider the following minimax excess risk over the class of linear estimators:

$$R_{\mathrm{lin}}(\mathcal{F}_\gamma) := \inf_{\widehat{f}:\mathrm{linear}} \sup_{f^\circ \in \mathcal{F}_\gamma} \mathrm{E}_{D_n}[\|\widehat{f} - f^\circ\|^2_{L_2(P_X)}],$$

where $\inf$ is taken over all linear estimators and $\mathrm{E}_{D_n}[\cdot]$ is taken with respect to the training data $D_n$. This expresses the best achievable worst case error over the class of linear estimators to estimate a function in $\mathcal{F}_\gamma$. To evaluate it, we additionally assume the following condition.

**Assumption 2.** *We assume that $\mu_m = m^{-2}$ and $a_m = \mu_m^{\alpha_1}$ $(m \in \mathbb{N})$ (and hence $c_\mu = 1$). There exists a monotonically decreasing sequence $(b_m)_{m=1}^\infty$ and $s \geq 3$ such that $b_m = \mu_m^{\alpha_2}$ $(\forall m)$ with $\alpha_2 > \gamma/2$ and $\sigma_m(u) = b_m^s \sigma(b_m^{-1} u)$ $(u \in \mathbb{R})$ where $\sigma$ is the sigmoid function: $\sigma(u) = 1/(1 + e^{-u})$.*

Intuitively, the parameter $s$ controls the "resolution" of each basis function $\sigma_m$, and the relation between parameter $\alpha_1$ and $\alpha_2$ controls the magnitude of coefficient for each basis $\sigma_m$. Note that the condition $s \geq 3$ ensures $\|\sigma_m\|_{1,3}$ is uniformly bounded and $0 < b_m \leq 1$ ensures $\|\sigma_m\|_\infty \leq 1$. Our main strategy to obtain the lower bound is to make use of the so-called *convex-hull argument*. That is, it is known that, for a function class $\mathcal{F}$, the minimax risk $R(\mathcal{F})$ over a class of linear estimators is identical to that for the convex hull of $\mathcal{F}$ (Hayakawa & Suzuki, 2020; Donoho et al., 1990):

$$R_{\mathrm{lin}}(\mathcal{F}) = R_{\mathrm{lin}}(\overline{\mathrm{conv}}(\mathcal{F})),$$

where $\mathrm{conv}(\mathcal{F}) = \{\sum_{i=1}^N \lambda_i f_i \mid f_i \in \mathcal{F}, \ \sum_{i=1}^N \lambda_i = 1, \ \lambda_i \geq 0, \ N \in \mathbb{N}\}$ and $\overline{\mathrm{conv}}(\cdot)$ is the closure of $\mathrm{conv}(\cdot)$ with respect to $L_2(P_X)$-norm. Intuitively, since the linear estimator is linear to the observations $(y_i)_{i=1}^n$ of outputs, a simple application of Jensen's inequality yields that its worst case error on the convex hull of the function class $\mathcal{F}$ does not increase compared with that on the original one $\mathcal{F}$ (see Hayakawa & Suzuki (2020) for the details). This indicates that the linear estimators cannot distinguish the original hypothesis class $\mathcal{F}$ and its convex hull. Therefore, if the class $\mathcal{F}$ is highly non-convex, then the linear estimators suffer from much slower convergence rate because its convex hull $\overline{\mathrm{conv}}(\mathcal{F})$ becomes much "fatter" than the original one $\mathcal{F}$. To make use of this argument, for each sample size $n$, we pick up appropriate $m_n$ and consider a subset generated by the basis function $\sigma_{m_n}$, i.e., $\mathcal{F}_\gamma^{(n)} := \{a_{m_n} \bar{w}_{2,m_n} \sigma_{m_n}(w_{1,m_n}^\top [x; 1]) \in \mathcal{F}_\gamma\}$. By applying the convex hull argument to this set, we obtain the relation $R_{\mathrm{lin}}(\mathcal{F}_\gamma) \geq R_{\mathrm{lin}}(\mathcal{F}_\gamma^{(n)}) = R_{\mathrm{lin}}(\overline{\mathrm{conv}}(\mathcal{F}_\gamma^{(n)}))$. Since $\mathcal{F}_\gamma^{(n)}$ is highly non-convex, its convex hull $\overline{\mathrm{conv}}(\mathcal{F}_\gamma^{(n)})$ is much larger than the original set $\mathcal{F}_\gamma^{(n)}$ and thus the minimax risk over the linear estimators would be much larger than that over all estimators including deep learning. More intuitively, linear estimators do not adaptively select the basis functions and thus they should prepare redundantly large class of basis functions to approximate functions in the target function class. The following theorem gives the lower bound of the minimax optimal excess risk over the class of linear estimators.

**Theorem 1.** *Suppose that $\mathrm{Var}(\epsilon) > 0$, $P_X$ is the uniform distribution on $[0,1]^d$, and Assumption 2 is satisfied. Let $\tilde{\beta} = \frac{\alpha_1 + (s+1)\alpha_2}{\alpha_2 - \gamma/2}$. Then for arbitrary small $\kappa' > 0$, we have that*

$$R_{\mathrm{lin}}(\mathcal{F}_\gamma) \gtrsim n^{-\frac{2\tilde{\beta}+d}{2\tilde{\beta}+2d}} n^{-\kappa'}. \tag{5}$$

The proof is in Appendix A. We utilized the Irie-Miyake integral representation (Irie & Miyake, 1988; Hornik et al., 1990) to show there exists a "complicated" function in the convex hull, and then we adopted the technique of Zhang et al. (2002) to show the lower bound. The lower bound is characterized by the decaying rate ($\alpha_1$) of $a_m$ relative to that ($\alpha_2$) of the scaling factor $b_m$. Indeed, the faster $a_m$ decays with increasing $m$, the faster the rate of the minimax lower bound becomes.

We can see that the minimax rate of linear estimators is quite sensitive to the dimension $d$. Actually, for relatively high dimensional settings, this lower bound becomes close to a slow rate $\Omega(1/\sqrt{n})$, which corresponds to the curse of dimensionality.

It has been pointed out that the sample complexity of kernel methods suffers from the curse of dimensionality while deep learning can avoid that by a tractable algorithms (e.g., Ghorbani et al. (2019); Bach (2017)). Among them, Ghorbani et al. (2019) showed that if the dimensionality $d$ is polynomial against $n$, then the excess risk of kernel methods is bounded away from 0 for all $n$. On the other hand, our analysis can be applied to *any* linear estimator including kernel methods, and it shows that even if the dimensionality $d$ is fixed, the convergence rate of their excess risk suffers from the curse of dimensionality. This can be accomplished thanks to a careful analysis of the rate of convergence. Bach (2017) derived an upper bound of the Rademacher complexity of the unit ball of the RKHS corresponding to a neural network model. However, it is just an upper bound and there is still a large gap from excess risk estimates. Allen-Zhu & Li (2019; 2020); Bai & Lee (2020); Chen et al. (2020) also analyzed a lower bound of sample complexity of kernel methods. However, their lower bound is not for the excess risk of the squared loss. Eventually, the sample complexities of all methods including deep learning take a form of $O(C/\sqrt{n})$ and dependency of coefficient $C$ to the dimensionality or other factors such as magnitude of residual components is compared. On the other hand, our lower bound properly involves the properties of squared loss such as strong convexity and smoothness and the bound shows the curse of dimensionality occurs even in the rate of convergence instead of just the coefficient. Finally, we would like to point out that several existing work (e.g., Ghorbani et al. (2019); Allen-Zhu & Li (2019)) considered a situation where the target function class changes as the sample size $n$ increases. However, our analysis reveals that separation between deep and shallow occurs even if the target function class $\mathcal{F}_\gamma$ is fixed.

### 4.2 UPPER BOUND FOR DEEP LEARNING

Here, we analyze the excess risk of deep learning trained by NGD and its algorithmic convergence rate. Our analysis heavily relies on the weak convergence of the discrete time gradient Langevin dynamics to the stationary distribution of the continuous time one (Eq. (4)). Under some assumptions, the continuous time dynamics has a stationary distribution (Da Prato & Zabczyk, 1992; Maslowski, 1989; Sowers, 1992; Jacquot & Royer, 1995; Shardlow, 1999; Hairer, 2002). If we denote the probability measure on $\mathcal{H}$ corresponding to the stationary distribution by $\pi_\infty$, then it is given by

$$\frac{d\pi_\infty}{d\nu_\beta}(W) \propto \exp(-\beta\widehat{\mathcal{L}}(f_W)),$$

where $\nu_\beta$ is the Gaussian measure in $\mathcal{H}$ with mean 0 and covariance $(\beta A)^{-1}$ (see Da Prato & Zabczyk (1996) for the rigorous definition of the Gaussian measure on a Hilbert space). Remarkably, this can be seen as *the Bayes posterior* for a prior distribution $\nu_\beta$ and a "log-likelihood" function $\exp(-\beta\widehat{\mathcal{L}}(W))$. Through this view point, we can obtain an excess risk bound of the solution $W_k$. The proofs of all theorems in this section are in Appendix B.

Under Assumption 1, the distribution of $W_k$ derived by the discrete time gradient Langevin synamics satisfies the following weak convergence property to the stationary distribution $\pi_\infty$. This convergence rate analysis depends on the techniques by Bréhier & Kopec (2016); Muzellec et al. (2020).

**Proposition 1.** *Assume Assumption 1 holds and $\beta > \eta$. Then, there exist spectral gaps $\Lambda_\eta^*$ and $\Lambda_0^*$ (defined in Eq. (10) of Appendix B.1) and a constant $C_0$ such that, for any $0 < a < 1/4$, the following convergence bound holds for almost sure observation $D_n$:*

$$|\mathrm{E}_{W_k}[\mathcal{L}(f_{W_k})|D_n] - \mathrm{E}_{W\sim\pi_\infty}[\mathcal{L}(f_W)|D_n]| \leq C_0\exp(-\Lambda_\eta^*\eta k) + C_1\frac{\sqrt{\beta}}{\Lambda_0^*}\eta^{1/2-a} =: \Xi_k, \quad (6)$$

*where $C_1$ is a constant depending only on $c_\mu, R, \alpha_1, C_\sigma, U, a$ (independent of $\eta, k, \beta, \lambda, n$).*

This proposition indicates that the expected risk of $W_k$ can be almost identical to that of the "Bayes posterior solution" obeying $\pi_\infty$ after sufficiently large iterations $k$ with sufficiently small step size $\eta$ even though $\widehat{\mathcal{L}}(f_W)$ is not convex. The definition of $\Lambda_\eta^*$ can be found in Eq. (10). We should note that its dependency on $\beta$ is exponential. Thus, if we take $\beta = \Omega(n)$, then the computational cost until a sufficiently small error could be exponential with respect to the sample size $n$. The same convergence holds also for finite dimensional one $W_k^{(M)}$ with a modified stationary distribution. The

constants appearing in the bound are independent of the model size $M$ (see the proof of Proposition 1 in Appendix B). In particular, the convergence can be guaranteed even if $W$ is infinite dimensional. This is quite different from usual finite dimensional analyses (Raginsky et al., 2017; Erdogdu et al., 2018; Xu et al., 2018) which requires exponential dependency on the dimension, but thanks to the regularization term, we can obtain the model size independent convergence rate. Xu et al. (2018) also analyzed a finite dimensional gradient Langevin dynamics and obtained a similar bound where $O(\eta)$ appears in place of the second term $\eta^{1/2-a}$ which corresponds to time discretization error. In our setting the regularization term is $\|W\|_{\mathcal{H}_1}^2 = \sum_m (\|w_{1,m}\|^2 + w_{2,m}^2)/\mu_m$ with $\mu_m \lesssim m^{-2}$, but if we employ $\|W\|_{\mathcal{H}_{p/2}}^2 = \sum_m (\|w_{1,m}\|^2 + w_{2,m}^2)/\mu_m^{p/2}$ for $p > 1$, then the time discretization error term would be modified to $\eta^{(p-1)/p-a}$ (Andersson et al., 2016). We can interpret the finite dimensional setting as the limit of $p \to \infty$ which leads to $\eta^{(p-1)/p} \to \eta$ that recovers the finite dimensional result ($O(\eta)$) as shown by Xu et al. (2018).

In addition to the above algorithmic convergence, we also have the following convergence rate for the excess risk bound of the finite dimensional solution $W_k^{(M)}$.

**Theorem 2.** *Assume Assumption 1 holds, assume $\eta < \beta \leq \min\{n/(2U^2), n\}$, and $0 < \gamma < 1/2 + \alpha_1$. Then, if the width satisfies $M \geq \min\left\{\lambda^{1/4\gamma(\alpha_1+1)}\beta^{1/2\gamma}, \lambda^{-1/2(\alpha_1+1)}, n^{1/2\gamma}\right\}$, the expected excess risk of $W_k$ is bounded as*

$$\mathrm{E}_{D^n}\Big[\mathrm{E}_{W_k^{(M)}}[\|f_{W_k^{(M)}} - f^\circ\|_{L_2(P_X)}^2 | D_n]\Big] \leq C \max\Big\{(\lambda\beta)^{\frac{1/\gamma}{1+1/2\gamma}} n^{-\frac{1}{1+1/2\gamma}}, \lambda^{-\frac{1}{2(\alpha_1+1)}}\beta^{-1}, \lambda^{\frac{\gamma}{1+\alpha_1}}\Big\} + \Xi_k,$$

*where $C$ is a constant independent of $n, \beta, \lambda, \eta, k$. In particular, if we set $\beta = \min\{n/(2U^2), n\}$ and $\lambda = \beta^{-1}$, then for $M \geq n^{1/2(\alpha_1+1)}$, we obtain*

$$\mathrm{E}_{D^n}\Big[\mathrm{E}_{W_k^{(M)}}[\|f_{W_k^{(M)}} - f^\circ\|_{L_2(P_X)}^2 | D_n]\Big] \lesssim n^{-\frac{\gamma}{\alpha_1+1}} + \Xi_k.$$

In addition to this theorem, if we further assume Assumption 2, we obtain a refined bound as follows.

**Corollary 1.** *Assume Assumptions 1 and 2 hold and $\eta < \beta$, and let $\beta = \min\{n/(2U^2), n\}$ and $\lambda = \beta^{-1}$. Suppose that there exists $0 \leq q \leq s - 3$ such that $0 < \gamma < 1/2 + \alpha_1 + q\alpha_2$. Then, the excess risk bound of $W_k^{(M)}$ for $M \geq n^{1/2(\alpha_1+q\alpha_2+1)}$ can be refined as*

$$\mathrm{E}_{D^n}\Big[\mathrm{E}_{W_k^{(M)}}[\|f_{W_k^{(M)}} - f^\circ\|_{L_2(P_X)}^2 | D_n]\Big] \lesssim n^{-\frac{\gamma}{\alpha_1+q\alpha_2+1}} + \Xi_k. \tag{7}$$

These theorem and corollary shows that the tractable NGD algorithm achieves a fast convergence rate of the excess risk bound. Indeed, if $q$ is chosen so that $\gamma > (\alpha_1 + q\alpha_2 + 1)/2$, then the excess risk bound converges faster than $O(1/\sqrt{n})$. Remarkably, the convergence rate is not affected by the input dimension $d$, which makes discrepancy from linear estimators. The bound of Theorem 2 is tightest when $\gamma$ is close to $1/2 + \alpha_1$ ($\gamma \approx 1/2 + \alpha_1 + 3\alpha_2$ for Corollary 1), and a smaller $\gamma$ yields looser bound. This relation between $\gamma$ and $\alpha_1$ reflects misspecification of the "prior" distribution. When $\gamma$ is small, the regularization $\lambda\|W\|_{\mathcal{H}_1}^2$ is not strong enough so that the variance of the posterior distribution becomes unnecessarily large for estimating the true function $f^\circ \in \mathcal{F}_\gamma$. Therefore, the best achievable bound can be obtained when the regularization is correctly specified. The analysis of fast rate is in contrast to some existing work (Allen-Zhu & Li, 2019; 2020; Li et al., 2020; Bai & Lee, 2020) that basically evaluated the Rademacher complexity. This is because we essentially evaluated a local Rademacher complexity instead.

## 4.3 COMPARISON BETWEEN LINEAR ESTIMATORS AND DEEP LEARNING

Here, we compare the convergence rate of excess risks between the linear estimators and the neural network method trained by NGD using the bounds obtained in Theorem 1 and Corollary 1 respectively. We write the lower bound (5) of the minimax excess risk of linear estimators as $R_{\mathrm{lin}}^*$ and the excess risk of the neural network approach (7) as $R_{\mathrm{NN}}^*$. To make the discussion concise, we consider a specific situation where $s = 3$, $\alpha_1 = \gamma = \frac{1}{4}\alpha_2$. In this case, $\tilde{\beta} = 17/3 \approx 5.667$, which gives

$$R_{\mathrm{lin}}^* \gtrsim n^{-\left(1+\frac{d}{2\tilde{\beta}+d}\right)^{-1}} n^{-\kappa'} \approx n^{-\left(1+\frac{d}{11.3+d}\right)^{-1}} n^{-\kappa'}.$$

On the other hand, by setting $q = 0$, we have

$$R^*_{\mathrm{NN}} \lesssim n^{-\frac{\alpha_1}{\alpha_1+1}} = n^{-\left(1+\frac{1}{\alpha_1}\right)^{-1}}.$$

Thus, as long as $\alpha_1 > 11.3/d + 1 \approx 2\tilde{\beta}/d + 1$, we have that

$$R^*_{\mathrm{lin}} \gtrsim R^*_{\mathrm{NN}}, \quad \text{and} \quad \lim_{n \to \infty} \frac{R^*_{\mathrm{NN}}}{R^*_{\mathrm{lin}}} = 0.$$

In particular, as $d$ gets larger, $R^*_{\mathrm{lin}}$ approaches to $\Omega(n^{-1/2})$ while $R^*_{\mathrm{NN}}$ is not affected by $d$ and it gets close to $O(n^{-1})$ as $\alpha_1$ gets larger. Moreover, the inequality $\alpha_1 > 11.3/d + 1$ can be satisfied by a relatively low dimensional setting; for example, $d = 10$ is sufficient when $\alpha_1 = 3$. As $\alpha_1$ becomes large, the model becomes "simpler" because $(a_m)_{m=1}^{\infty}$ decays faster. However, the linear estimators cannot take advantage of this information whereas deep learning can. From the convex hull argument, this discrepancy stems from the non-convexity of the model. We also note that the superiority of deep learning is shown *without* sparse regularization while several existing work showed favorable estimation property of deep learning though sparsity inducing regularization (Bach, 2017; Chizat, 2019; Hayakawa & Suzuki, 2020). However, our analysis indicates that sparse regularization is not necessarily as long as the model has non-convex geometry, i.e., sparsity is just one sufficient condition for non-convexity but not a necessarily condition. The parameter setting above is just a sufficient condition and the lower bound $R^*_{\mathrm{lin}}$ would not be tight. The superiority of deep learning would hold in much wider situations.

## 5 CONCLUSION

In this paper, we studied excess risks of linear estimators, as a representative of shallow methods, and a neural network estimator trained by a noisy gradient descent where the model is fixed and no sparsity inducing regularization is imposed. Our analysis revealed that deep learning can outperform any linear estimator even for a relatively low dimensional setting. Essentially, non-convexity of the model induces this difference and the curse of dimensionality for linear estimators is a consequence of a fact that the geometry of the model becomes more "non-convex" as the dimension of input gets higher. All derived bounds are fast rate because the analyses are about the excess risk with the squared loss, which made it possible to compare the rate of convergence. The fast learning rate of the deep learning approach is derived through the fact that the noisy gradient descent behaves like a Bayes estimator with model size independent convergence rate.

### ACKNOWLEDGMENTS

TS was partially supported by JSPS Kakenhi (18K19793, 18H03201, and 20H00576), Japan Digital Design and JST-CREST.

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

# A  PROOF OF THEOREM 1

We basically combine the "convex hull argument" and the minimax optimal rate analysis for linear estimators developed by Zhang et al. (2002).

Zhang et al. (2002) essentially showed the following statement in their Theorem 1.

**Proposition 2** (Theorem 1 of Zhang et al. (2002))**.** *Let $\mu$ be the Lebesgue measure. Suppose that the space $\Omega$ has even partition $\mathcal{A}$ such that $|\mathcal{A}| = 2^K$ for an integer $K \in \mathbb{N}$, each $A$ has equivalent measure $\mu(A) = 2^{-K}$ for all $A \in \mathcal{A}$, and $\mathcal{A}$ is indeed a partition of $\Omega$, i.e., $\cup_{A \in \mathcal{A}} = \Omega$, $A \cap A' = \emptyset$ for $A, A' \in \Omega$ and $A \neq A'$. Then, if $K$ is chosen as $n^{-\gamma_1} \leq 2^{-K} \leq n^{-\gamma_2}$ for constants $\gamma_1, \gamma_2 > 0$ that are independent of $n$, then there exists an event $\mathcal{E}$ such that, for a constant $C' > 0$,*

$$P(\mathcal{E}) \geq 1 + o(1) \text{ and } |\{x_i \mid x_i \in A \, (i \in \{1, \ldots, n\})\}| \leq C'n/2^K \quad (\forall A \in \mathcal{A}).$$

*Moreover, suppose that, for a class $\mathcal{F}^\circ$ of functions on $\Omega$, there exists $\Delta > 0$ that satisfies the following conditions:*

1. *There exists $F > 0$ such that, for any $A \in \mathcal{A}$, there exists $g \in \mathcal{F}^\circ$ that satisfies $g(x) \geq \frac{1}{2}\Delta F$ for all $x \in A$,*

2. *There exists $K'$ and $C'' > 0$ such that $\frac{1}{n}\sum_{i=1}^{n} g(x_i)^2 \leq C''\Delta^2 2^{-K'}$ for any $g \in \mathcal{F}^\circ$ on the event $\mathcal{E}$.*

*Then, there exists a constant $F_1$ such that at least one of the following inequalities holds:*

$$\frac{F^2}{4F_1C''}\frac{2^{K'}}{n} \leq R_{\text{lin}}(\mathcal{F}^\circ), \tag{8a}$$

$$\frac{F^3}{32}\Delta^2 2^{-K} \leq R_{\text{lin}}(\mathcal{F}^\circ), \tag{8b}$$

*for sufficiently large $n$.*

Before we show the main assertion, we prepare some additional lemmas. For a sigmoid function $\sigma$, let $\tilde{\mathcal{F}}_{C,\tau}^{(\sigma)} := \{x \in \mathbb{R}^d \mapsto a\sigma(\tau(w^\top x + b))) \mid |a| \leq 2C, \|w\| \leq 1, |b| \leq 2 \, (a, b \in \mathbb{R}, \, w \in \mathbb{R}^d)\}$ for $C > 0, \tau > 0$.

**Lemma 1.** *Let $\psi(x) = \frac{1}{2}(\sigma(x+1) - \sigma(x-1))$ and $\hat{\psi}$ be its Fourier transform: $\hat{\psi}(\omega) := (2\pi)^{-1}\int e^{-\mathrm{i}\omega x}\psi(x)\mathrm{d}x$. Let $h > 0$ and $D_w > 0$. Then, by setting $\tau = h^{-1}(2\sqrt{d}+1)D_w$ and $C = \frac{(2\sqrt{d}+1)D_w}{\pi h|\hat{\psi}(1)|}$, the Gaussian RBF kernel can be approximated by*

$$\inf_{\check{g} \in \overline{\text{conv}}(\tilde{\mathcal{F}}_{C,\tau}^{(\sigma)})} \sup_{x \in [0,1]^d} \left|\check{g}(x) - \exp\left(-\frac{\|x - c\|^2}{2h^2}\right)\right|$$

$$\leq \frac{4}{|2\pi\hat{\psi}(1)|}\left[C_d D_w^{2(d-2)}\exp(-D_w^2/2) + \exp(-D_w)\right]$$

*for any $c \in [0,1]^d$, where $C_d$ is a constant depending only on $d$. In particular, the right hand side is $O(\exp(-n^\kappa))$ if $D_w = n^\kappa$.*

*Proof of Lemma 1.* Let $\psi_h(x) = \psi(h^{-1}x)$. Suppose that, for $f \in L_1(\mathbb{R}^d)$, its Fourier transform $\hat{f}(\omega) = (2\pi)^{-d}\int e^{-\mathrm{i}\omega^\top x}f(x)\mathrm{d}x \, (\omega \in \mathbb{R}^d)$ gives

$$\int_{\mathbb{R}^d} \exp(\mathrm{i}w^\top x)\hat{f}(w)\mathrm{d}w = f(x),$$

for every $x \in \mathbb{R}^d$[1]. Then the Irie-Miyake itegral representation (Irie & Miyake (1988); see also the proof of Theorem 3.1 in Hornik et al. (1990)) gives

$$f(x) = \int_{a \in \mathbb{R}^d}\int_{b \in \mathbb{R}} \psi(a^\top x + b)\mathrm{d}\nu(a, b) \quad \text{(a.e.)},$$

---

[1] If $\hat{f}$ is integrable, this inversion formula holds for almost every $x \in \mathbb{R}^d$ (Rudin, 1987). However, we assume a stronger condition that it holds for every $x \in \mathbb{R}^d$.

where $\mathrm{d}\nu(a, b)$ is given by

$$\mathrm{d}\nu(a, b) = \mathrm{Re}\left(\frac{|\omega|^d e^{-iwb}}{2\pi\hat{\psi}(\omega)}\right)\hat{f}(wa)\mathrm{d}a\mathrm{d}b$$

for any $\omega \neq 0$. Since the characteristic function of the multivariate normal distribution gives that

$$\int_{\mathbb{R}^d}\exp(iw^\top(x-c))\underbrace{\sqrt{\frac{h^{2d}}{(2\pi)^d}}\exp\left(-\frac{h^2\|w\|^2}{2}\right)}_{=\hat{f}(w)}\mathrm{d}w = \exp\left(-\frac{\|x-c\|^2}{2h^2}\right) =: f(x) \ (\forall x \in \mathbb{R}^d),$$

we have that

$$\exp\left(-\frac{\|x-c\|^2}{2h^2}\right) =$$

$$\int_{a\in\mathbb{R}^d}\int_{b\in\mathbb{R}}\psi_h(a^\top(x-c)+b)\mathrm{Re}\left(\frac{e^{-iwb}}{2\pi\hat{\psi}_h(\omega)}\right)\sqrt{\frac{|\omega h|^{2d}}{(2\pi)^d}}\exp\left(-\frac{(\omega h)^2\|a\|^2}{2}\right)\mathrm{d}a\mathrm{d}b,$$

for all $x \in \mathbb{R}^d$. Since $\psi_h(\cdot) = \psi(h^{-1}\cdot)$ and $\hat{\psi}_h(\cdot) = h\hat{\psi}(h\cdot)$ by its definition, the right hand side is equivalent to

$$\int_{a\in\mathbb{R}^d}\int_{b\in\mathbb{R}}\psi(h^{-1}[a^\top(x-c)+b])\mathrm{Re}\left(\frac{e^{-iwb}}{2\pi h\hat{\psi}(h\omega)}\right)\sqrt{\frac{|\omega h|^{2d}}{(2\pi)^d}}\exp\left(-\frac{(\omega h)^2\|a\|^2}{2}\right)\mathrm{d}a\mathrm{d}b.$$

Here, we set $\omega = h^{-1}$. Let $N_{\sigma^2}$ be the probability measure corresponding to the multivariate normal with mean 0 and covariance $\sigma^2\mathrm{I}$, and let $A_D := \{w \in \mathbb{R}^d \mid \|w\| \leq D\}$. Let $D_a > 0$ and $D_b = D_a(\sqrt{2d}+1)$, and define

$$f_{D_a}(x) := \frac{1}{2D_b N_1(A_{D_a})}\int_{\|a\|\leq D_a, |b|\leq D_b}\psi(h^{-1}[a^\top(x-c)+b])\mathrm{Re}\left(\frac{e^{-ib/h}}{2\pi h\hat{\psi}(1)}\right)\times$$

$$\sqrt{\frac{1}{(2\pi)^d}}\exp\left(-\frac{\|a\|^2}{2}\right)\mathrm{d}a\mathrm{d}b.$$

Then, we can see that, for any $x \in [0, 1]^d$, it holds that

$$\left|\frac{1}{2D_b N_1(A_{D_a})}f(x) - f_{D_a}(x)\right|$$

$$\leq \frac{1}{2D_b N_1(A_{D_a})|2\pi h\hat{\psi}(1)|}\left[N_1(A_{D_a}^c)\int 2\exp(-h^{-1}|x|)\mathrm{d}x + \int_{|b|>D_b}2\exp(-[h^{-1}(|b|-2\sqrt{d}D_a)])\mathrm{d}b\right]$$

$$\leq \frac{1}{2D_b N_1(A_{D_a})|2\pi h\hat{\psi}(1)|}\left[4hN_1(A_{D_a}^c) + 4h\exp(-D_a)\right]$$

$$= \frac{4h}{2D_b N_1(A_{D_a})|2\pi h\hat{\psi}(1)|}\left[C_d D_a^{2(d-2)}\exp(-D_a^2/2) + \exp(-D_a)\right],$$

where $C_d > 0$ is a constant depending on only $d$, and we used $|a^\top(x-c)+b| \geq |b| - |a^\top(x-c)| \geq |b| - 2\sqrt{d}D_a$ and $\psi(x) \leq 2\exp(-|x|)$. Note that if $D_a = n^\kappa$, then the right hand side is $O(h\exp(-n^\kappa))$. Therefore, since $N_1(A_{D_a}) \leq 1$, by setting $\tau = h^{-1}D_b$, $C = \frac{D_b}{\pi h|\hat{\psi}(1)|}$, we have that

$$\inf_{\check{g}\in\overline{\mathrm{conv}}(\tilde{\mathcal{F}}_{C,\tau}^{(\sigma)})}\sup_{x\in[0,1]^d}\left|\check{g}(x) - \exp\left(-\frac{\|x-c\|^2}{2h^2}\right)\right|$$

$$\leq \frac{4}{|2\pi\hat{\psi}(1)|}\left[C_d D_a^{2(d-2)}\exp(-D_a^2/2) + \exp(-D_a)\right].$$

Hence, by rewriting $D_w \leftarrow D_a$, we obtain the assertion. As noted above, the right hand is $O(\exp(-n^\kappa))$ if $D_a = n^\kappa$. □

*Proof of Theorem 1.* For a sample size $n$, we fix $m_n$ which will be determined later and use Proposition 2 with $\mathcal{F}^\circ = \mathcal{F}_\gamma^{(n)}$. If $w_{2,m_n} = b\sqrt{\mu_{m_n}^\gamma/2}$ with $|b| \leq 1$ and $w_{1,m} = \mu_{m_n}^{\gamma/2}[u; -u^\top c]/(\sqrt{2(d+1)})$ for $u \in \mathbb{R}^d$ such that $\|u\| \leq 1$ and $c \in [0,1]^d$, then $\|(w_{1,m_n}, w_{2,m_n})\|^2 \leq \mu_{m_n}^\gamma(1/2 + (1 + |u^\top c|^2)/2(d+1)) \leq \mu_{m_n}^\gamma$. Therefore, $\tilde{\varphi}_{u,c}(x) = a_{m_n}\bar{w}_{2,m_n}\sigma_{m_n}(w_{1,m_n}^\top[x;1]) = \mu_{m_n}^{\alpha_1}(b\mu_{m_n}^{\gamma/2}/\sqrt{2})\mu_{m_n}^{s\alpha_2}\sigma\left(\mu_{m_n}^{-\alpha_2+\gamma/2}u^\top(x-c)/\sqrt{2(d+1)}\right) \in \mathcal{F}_\gamma^{(n)} \subset \mathcal{F}_\gamma$ for all $b \in \mathbb{R}$ with $|b| \leq 1$, $u \in \mathbb{R}^d$ with $\|u\| \leq 1$, and $c \in [0,1]^d$. In other words, $\mu_{m_n}^{\alpha_1+\gamma/2+s\alpha_2}(2C)^{-1}\mathcal{F}_{C,\tau}^{(\sigma)} \subset \mathcal{F}_\gamma^{(n)}$ for any $C > 0$ and $\tau = \frac{1}{\sqrt{2(d+1)}}\mu_{m_n}^{-\alpha_2+\gamma/2}$.

Therefore, by setting $C = (\sqrt{2d}+1)D_w/(\pi h|\hat{\psi}(1)|)$ for $D_w > 0$, Lemma 1 yields that for any $c \in [0,1]^d$ and given $h > 0$, there exists $g \in \overline{\mathrm{conv}}(\mathcal{F}_\gamma^{(n)})$ such that

$$\left\|\mu_{m_n}^{\alpha_1+\gamma/2+s\alpha_2}\left(\frac{2(\sqrt{2d}+1)D_w}{\pi h|\hat{\psi}(1)|}\right)^{-1}\exp\left(-\frac{\|\cdot - c\|^2}{2h^2}\right) - g\right\|_\infty$$

$$\leq \mu_{m_n}^{\alpha_1+\gamma/2+s\alpha_2}\left(\frac{2(\sqrt{2d}+1)D_w}{\pi h|\hat{\psi}(1)|}\right)^{-1}\frac{4}{|2\pi\hat{\psi}(1)|}\left[C_d D_w^{2(d-2)}\exp(-D_w^2/2) + \exp(-D_w)\right]$$

$$= \mu_{m_n}^{\alpha_1+\gamma/2+s\alpha_2}\frac{h}{(\sqrt{2d}+1)D_w}\left[C_d D_a^{2(d-2)}\exp(-D_w^2/2) + \exp(-D_w)\right].$$

We let $D_w = n^\kappa$ for any $\kappa > 0$ and choose $\mu_{m_n}$ as $\tau \simeq \mu_{m_n}^{-\alpha_2+\gamma/2} = D_w h^{-1} = h^{-1}n^\kappa$. We write $\Delta := \mu_{m_n}^{\alpha_1+\gamma/2+s\alpha_2}(2C)^{-1} \simeq h^{\frac{\alpha_1+s\alpha_2+\gamma/2}{\alpha_2-\gamma/2}+1}n^{-\kappa(\frac{\alpha_1+s\alpha_2+\gamma/2}{\alpha_2-\gamma/2}+1)}$. Then, it holds that

$$\left\|\Delta\exp\left(-\frac{\|\cdot - c\|^2}{2h^2}\right) - g\right\|_\infty \lesssim \Delta\exp(-n^\kappa). \tag{9}$$

Here, we set $h$ as $h = 2^{-k}$ with a positive integer $k$. Accordingly, we define a partition $\mathcal{A}$ of $\Omega$ so that any $A \in \mathcal{A}$ can be represented as $A = [2^{-k}j_1, 2^{-k}(j_1+1)] \times \cdots \times [2^{-k}j_d, 2^{-k}(j_d+1)]$ by non-negative integers $0 \leq j_i \leq 2^k - 1$ $(i = 1, \ldots, d)$. Note that $|\mathcal{A}| = 2^{dk} = h^{-d}$.

For each $A \in \mathcal{A}$, we define $c_A$ as $c_A = (2^{-k}(j_1 + 1/2), \ldots, 2^{-k}(j_d + 1/2))^\top$ where $(j_1, \ldots, j_d)$ is a set of indexes that satisfies $A = [2^{-k}j_1, 2^{-k}(j_1+1)] \times \cdots \times [2^{-k}j_d, 2^{-k}(j_d+1)]$. For each $A \in \mathcal{A}$, we define $g_A \in \overline{\mathrm{conv}}(\mathcal{F}_\gamma^{(n)})$ as a function that satisfies Eq. (9) for $c = c_A$.

Now, we apply Proposition 2 with $\mathcal{F}^\circ = \overline{\mathrm{conv}}(\mathcal{F}_\gamma^{(n)})$ and $K = K' = dk$. Let $R^* := R_{\mathrm{lin}}(\overline{\mathrm{conv}}(\mathcal{F}_\gamma^{(n)}))$. First, we can see that there exits a constant $F > 0$ such that

$$g_A(x) \geq F\Delta \quad (\forall x \in A),$$

where we used $\exp(-n^\kappa) \ll 1$.

Second, in the event $\mathcal{E}$ introduced in the statement of Proposition 2, there exists $C$ such that $|\{i \in \{1, \ldots, n\} \mid x_i \in A'\}| \leq Cn/2^{-dk}$ for all $A' \in \mathcal{A}$. In this case, we can check that

$$\frac{1}{n}\sum_{i=1}^n\left[\Delta\exp\left(-\frac{\|x_i - c_A\|^2}{2h^2}\right)\right]^2 \lesssim \Delta^2 h^d = \Delta^2 2^{-kd},$$

by the uniform continuity of the Gaussian RBF. Therefore, we also have

$$\frac{1}{n}\sum_{i=1}^n g_A(x_i)^2 \leq \frac{2}{n}\sum_{i=1}^n\left[\Delta\exp\left(-\frac{\|x_i - c_A\|^2}{2h^2}\right)\right]^2 + c\Delta^2\exp(-2n^\kappa)$$

$$\lesssim \Delta^2(h^d + \exp(-2n^\kappa)),$$

where $c > 0$ is a constant. Thus, as long as $h$ is polynomial to $n$ like $h = \Theta(n^{-a})$, the right hand side is $O(\Delta^2 h^d)$.

Now, if we write

$$\tilde{\beta} = \frac{\alpha_1 + s\alpha_2 + \gamma/2}{\alpha_2 - \gamma/2} + 1 = \frac{\alpha_1 + (s+1)\alpha_2}{\alpha_2 - \gamma/2},$$

then we have $\Delta \simeq h^{\tilde{\beta}} n^{-\kappa\tilde{\beta}}$ by its definition.

Here, we choose $k$ as a maximum integer that satisfies $\frac{F^3}{32}\Delta^2 2^{-dk} > R^*$. In this situation, it holds that

$$h^{2\tilde{\beta}+d} n^{-2\kappa\tilde{\beta}} \simeq R^*.$$

Since Eq. (8b) is not satisfied, Eq. (8a) must hold, and hence we have

$$n^{-1}h^{-d} \lesssim R^* \simeq h^{2\tilde{\beta}+d} n^{-2\kappa\tilde{\beta}}$$

$$\Rightarrow \quad h \simeq n^{-\frac{1-2\kappa\tilde{\beta}}{2\tilde{\beta}+2d}}.$$

Therefore, we obtain that

$$R^* \gtrsim n^{-\frac{2\tilde{\beta}+d}{2\tilde{\beta}+2d}} n^{-\frac{2\kappa d\tilde{\beta}}{2\tilde{\beta}+2d}}$$

$$\geq n^{-\frac{2\tilde{\beta}+d}{2\tilde{\beta}+2d}} n^{-\kappa'},$$

by setting $\kappa' = \kappa\frac{2d\tilde{\beta}}{2\tilde{\beta}+2d}$. This gives the assertion. □

# B  PROOFS OF PROPOSITION 1, THEOREM 2 AND COROLLARY 1

Proposition 1, Theorem 2 and Corollary 1 can be shown by using Propositions 3 and 4 given in Appendix B.1 shown below.

Let $T^\alpha W = (\mu_m^\alpha w_{1,m}, \mu_m^\alpha w_{2,m})_{m=1}^\infty$ for $W = (w_{1,m}, w_{2,m})_{m=1}^\infty$ for $\alpha > 0$, and let us consider a model $h_W := f_{T^{-\alpha/2}W}$. Then, the training error can be rewritten as

$$\widehat{\mathcal{L}}(f_W) = \widehat{\mathcal{L}}(h_{T^{\alpha/2}W}).$$

For notational simplicity, we let $\widehat{\mathcal{L}}(W) := \widehat{\mathcal{L}}(f_W)$.

Let $\mathcal{H}^{(M)}$ be $\{W^{(M)} = (w_{1,m}, w_{2,m})_{m=1}^M \mid w_{1,m} \in \mathbb{R}^{d+1}, w_{2,m} \in \mathbb{R}, 1 \leq m \leq M\}$ and $\iota : \mathcal{H}^{(M)} \to \mathcal{H}$ be the zero padding of $W^{(M)}$, that is, $\iota(W^{(M)}) = (w'_{1,m}, w'_{2,m})_{m=1}^\infty \in \mathcal{H}$ satisfies $w'_{1,m} = w_{1,m}$, $w'_{2,m} = w_{2,m}$ ($m \leq M$) and $w'_{1,m} = 0$, $w'_{2,m} = 0$ ($m > M$). Moreover, we define $\iota^* : \mathcal{H} \to \mathcal{H}^{(M)}$ as the map that extracts first $M$ components. By abuse of notation, we write $f_{W^{(M)}}$ for $W^{(M)} \in \mathcal{H}^{(M)}$ to indicate $f_{\iota(W^{(M)})}$. Finally, let $A^{(M)} : \mathcal{H}^{(M)} \to \mathcal{H}^{(M)}$ be a linear operator such that $A^{(M)}W^{(M)} = \iota^*(A\iota(W^{(M)}))$, which is just a truncation of $A$. Similarly, let $T_M^a W^{(M)}$ for $W^{(M)} \in \mathcal{H}^{(M)}$ be the operator corresponding to $T^a W$ for $W \in \mathcal{H}$, i.e., $T_M^a W^{(M)} = \iota^*(T^a\iota(W^{(M)}))$.

## B.1  AUXILIARY LEMMAS

First, we show some key propositions to show the main results. To do so, we utilize the result by Muzellec et al. (2020) and Suzuki (2020).

**Assumption 3.**

(i) *There exists a constant $c_\mu$ such that $\mu_m \leq c_\mu m^{-2}$.*

(ii) *There exist $B, U > 0$ such that the following two inequalities hold for some $a \in (1/4, 1)$ almost surely:*

$$\|\nabla\widehat{\mathcal{L}}(W)\|_{\mathcal{H}} \leq B \ (\forall W \in \mathcal{H}),$$

$$\|\nabla\widehat{\mathcal{L}}(W) - \nabla\widehat{\mathcal{L}}(W')\|_{\mathcal{H}} \leq L\|W - W'\|_{\mathcal{H}_{-a}} \ (\forall W, W' \in \mathcal{H}).$$

(iii) *For any data $D_n$, $\widehat{\mathcal{L}}$ is three times differentiable. Let $\nabla^3\widehat{\mathcal{L}}(W)$ be the third-order derivative of $\widehat{\mathcal{L}}(W)$. This can be identified with a third-order linear form and $\nabla^3\widehat{\mathcal{L}}(W)\cdot(h,k)$ denotes the Riesz representor of $l \in \mathcal{H} \mapsto \nabla^3\widehat{\mathcal{L}}(W) \cdot (h,k,l)$. There exists $\alpha' \in [0,1), C_{\alpha'} \in (0,\infty)$ such that $\forall W, h, k \in \mathcal{H}, \|\nabla^3\widehat{\mathcal{L}}(W) \cdot (h,k)\|_{\mathcal{H}_{-\alpha'}} \leq C_{\alpha'}\|h\|_{\mathcal{H}}\|k\|_{\mathcal{H}}, \;\; \|\nabla^3\widehat{\mathcal{L}}(W) \cdot (h,k)\|_{\mathcal{H}} \leq C_{\alpha'}\|h\|_{\mathcal{H}_{\alpha'}}\|k\|_{\mathcal{H}}$ (a.s.).*

**Remark 1.** *In the analysis of Bréhier & Kopec (2016); Muzellec et al. (2020); Suzuki (2020), Assumption 3-(iii) is imposed for any finite dimensional projection $\mathcal{L}(W^{(M)})$ as a function on $\mathcal{H}^{(M)}$) for all $M \geq 1$ instead of $\mathcal{L}(W)$ as a function of $\mathcal{H}$. However, the condition on $\mathcal{L}(W)$ gives a sufficient condition for any finite dimensional projection in our setting. Thus, we employed the current version.*

**Assumption 4.** *For the loss function $\ell(y, f(x)) = (y - f(x))^2$, the following conditions holds:*

(i) *There exists $C > 0$ such that for any $f_W$ ($W \in \mathcal{H}$), it holds that*

$$\mathrm{E}_{X,Y}[(\ell(Y, f_W(X)) - \ell(Y, f^*(X)))^2] \leq C(\mathcal{L}(f_W) - \mathcal{L}(f^*)).$$

(ii) *$\beta > 0$ is chosen so that, for any $h : \mathbb{R}^d \to \mathbb{R}$ and $x \in \mathrm{supp}(P_X)$, it holds that*

$$\mathrm{E}_{Y|X=x}\big[\exp\big(-\tfrac{\beta}{n}(\ell(Y, h(x)) - \ell(Y, f^*(x)))\big)\big] \leq 1.$$

(iii) *There exists $L_h > 0$ such that $\|\nabla_W\ell(Y, h_W(X)) - \nabla_W\ell(Y, h_{W'}(X))\|_{\mathcal{H}} \leq L_h\|W - W'\|_{\mathcal{H}}$ ($\forall W, W' \in \mathcal{H}$) almost surely.*

(iv) *There exists $C_h$ such that $\|h_W - h_{W'}\|_\infty \leq C_h\|W - W'\|_{\mathcal{H}}$ ($W, W' \in \mathcal{H}$).*

**Proposition 3.** *Assume Assumption 3 holds and $\beta > \eta$. Suppose that $\exists \bar{R} > 0, 0 \leq \ell(Y, f_W(X)) \leq \bar{R}$ for any $W \in \mathcal{H}$ (a.s.). Let $\rho = \frac{1}{1+\lambda\eta/\mu_1}$ and $b = \frac{\mu_1}{\lambda}B + \frac{c_\mu}{\beta\lambda}$. Accordingly, let $\bar{b} = \max\{b, 1\}$, $\kappa = \bar{b} + 1$ and $\bar{V} = 4\bar{b}/(\sqrt{(1+\rho^{1/\eta})/2} - \rho^{1/\eta})$. Then, the spectral gap of the dynamics is given by*

$$\Lambda_\eta^* = \frac{\min\left(\frac{\lambda}{2\mu_1}, \frac{1}{2}\right)}{4\log(\kappa(\bar{V}+1)/(1-\delta))}\delta \tag{10}$$

*where $0 < \delta < 1$ is a real number satisfying $\delta = \Omega(\exp(-\Theta(\mathrm{poly}(\lambda^{-1})\beta)))$. We define $\Lambda_0^* = \lim_{\eta\to 0}\Lambda_\eta^*$ (i.e., $\bar{V}$ is replaced by $4\bar{b}/(\sqrt{(1+\exp(-\frac{\lambda}{\mu_1}))/2} - \exp(-\frac{\lambda}{\mu_1})))$. We also define $C_{W_0} = \kappa[\bar{V} + 1] + \frac{\sqrt{2}(\bar{R}+b)}{\sqrt{\delta}}$. Then, for any $0 < a < 1/4$, the following convergence bound holds for almost sure observation $D_n$: for either $L = \mathcal{L}$ or $L = \widehat{\mathcal{L}}$,*

$$|\mathrm{E}_{W_k}[L(W_k)|D_n] - \mathrm{E}_{W\sim\pi_\infty}[L(W)|D_n]| \tag{11}$$

$$\leq C_1\left[C_{W_0}\exp(-\Lambda_\eta^*\eta k) + \frac{\sqrt{\beta}}{\Lambda_0^*}\eta^{1/2-a}\right] = \Xi'_k, \tag{12}$$

*where $C_1$ is a constant depending only on $c_\mu, B, L, C_{\alpha'}, a, \bar{R}$ (independent of $\eta, k, \beta, \lambda$).*

**Proposition 4.** *Assume that Assumptions 3 and 4 hold. Let $\tilde{\alpha} := 1/\{2(\alpha + 1)\}$ for a given $\alpha > 0$ and $\theta$ be an arbitrary real number satisfying $0 < \theta < 1 - \tilde{\alpha}$. Assume that the true function $f^\circ$ can be represented by $h_{W^*} = f^\circ$ for $W^* \in \mathcal{H}_{\theta(\alpha+1)}$. Then, if $M \geq \min\{\lambda^{\tilde{\alpha}/2[\theta(\alpha+1)]}\beta^{1/2[\theta(\alpha+1)]}, \lambda^{-1/2(\alpha+1)}, n^{1/2[\theta(\alpha+1)]}\}$, the expected excess risk is bounded by*

$$\mathrm{E}_{D^n}\left[\mathrm{E}_{W_k^{(M)}}[\mathcal{L}(h_{T_M^{\alpha/2}W_k^{(M)}})|D_n] - \mathcal{L}(f^\circ)\right]$$

$$\leq C\max\left\{(\lambda\beta)^{\frac{2\tilde{\alpha}/\theta}{1+\tilde{\alpha}/\theta}}n^{-\frac{1}{1+\tilde{\alpha}/\theta}}, \lambda^{-\tilde{\alpha}}\beta^{-1}, \lambda^\theta, 1/n\right\} + \Xi'_k, \tag{13}$$

*where $C$ is a constant independent of $n, \beta, \lambda, \eta, k$.*

*Proof.* Repeating the same argument in Proposition 1 and using the same notation, Proposition 3 gives

$$|\mathrm{E}_{W_k^{(M)}}[\mathcal{L}(W_k^{(M)})|D_n] - \mathrm{E}_{W\sim\pi_\infty^{(M)}}[\mathcal{L}(W)|D_n]| \leq \Xi'_k,$$

for any $1 \leq M \leq \infty$. Therefore, we just need to bound the following quantity:
$$\left| \mathrm{E}_{D^n} \left[ \mathrm{E}_{W^{(M)} \sim \pi_\infty^{(M)}} [\mathcal{L}(h_{T_M^{\alpha/2} W^{(M)}}) | D_n] \right] - \mathcal{L}(f^\circ) \right|.$$

We define $\|W^{(M)}\|_{\mathcal{H}^{(M)}} := \|\iota^*(W^{(M)})\|_{\mathcal{H}}$ for $W^{(M)} \in \mathcal{H}^{(M)}$. For $a > 0$, we define $\mathcal{H}_a^{(M)}$ be the projection of $\mathcal{H}_a$ to the first $M$ components, $\mathcal{H}_a^{(M)} = \{\iota(W) \mid W \in \mathcal{H}_a\}$, and we define $\|W^{(M)}\|_{\mathcal{H}_a^{(M)}} := \|\iota^*(W^{(M)})\|_{\mathcal{H}_a}$ (note that since $\mathcal{H}_a^{(M)}$ is a finite dimensional linear space, it is same as $\mathcal{H}$ as a set). Let $\nu_\beta^{(M)}$ be the Gaussian measure on $\mathcal{H}^{(M)}$ with mean 0 and covariance $(\beta A^{(M)})^{-1}$, and $\tilde{\nu}_\beta^{(M)}$ be the Gaussian measure corresponding to the random variable $T_M^{\alpha/2} W^{(M)}$ with $W^{(M)} \sim \nu_\beta^{(M)}$. Let the concentration function be

$$\phi_{\beta,\lambda}^{(M)}(\epsilon) := \inf_{\substack{W \in \mathcal{H}_{\alpha+1}^{(M)}: \\ \mathcal{L}(h_W) - \mathcal{L}(f^\circ) \leq \epsilon^2}} \beta\lambda\|W\|_{\mathcal{H}_{\alpha+1}^{(M)}}^2 - \log \tilde{\nu}_\beta^{(M)}(\{W \in \mathcal{H}^{(M)} : \|W\|_{\mathcal{H}^{(M)}} \leq \epsilon\}) + \log(2),$$

where, if there does not exist $W \in \mathcal{H}_{\alpha+1}^{(M)}$ that satisfies the condition inf, then we define $\phi_{\beta,\lambda}^{(M)}(\epsilon) = \infty$, then Let $\epsilon^* > 0$ be

$$\epsilon^* := \max\{\inf\{\epsilon > 0 \mid \phi_{\beta,\lambda}(\epsilon) \leq \beta\epsilon^2\}, 1/n\}.$$

Then, Suzuki (2020) showed the following bound:

$$\left| \mathrm{E}_{D^n} \left[ \mathrm{E}_{W^{(M)} \sim \pi_\infty^{(M)}} [\mathcal{L}(h_{T_{(M)}^{\alpha/2} W^{(M)}}) | D_n] - \mathcal{L}(f^\circ) \right] \right|$$
$$\leq C \max\left\{ \epsilon^{*2}, \left(\tfrac{\beta}{n}\epsilon^{*2} + n^{-\frac{1}{1+\tilde{\alpha}/\theta}}(\lambda\beta)^{\frac{2\tilde{\alpha}/\theta}{1+\tilde{\alpha}/\theta}}\right), \tfrac{1}{n} \right\}. \tag{14}$$

They also showed that, for $M = \infty$, it holds that

$$\epsilon^{*2} \lesssim \max\left\{ (\lambda\beta)^{-\tilde{\alpha}} \beta^{-(1-\tilde{\alpha})}, \lambda^\theta, n^{-1} \right\} = \max\left\{ \lambda^{-\tilde{\alpha}} \beta^{-1}, \lambda^\theta, n^{-1} \right\}.$$

Substituting this bound of $\epsilon^*$ to Eq. (14), we obtain Eq. (13) for $M = \infty$. Moreover, in their proof, if $M \geq (\epsilon^*)^{-1/[\theta(\alpha+1)]}$, then

$$\inf_{\substack{W \in \mathcal{H}_{\alpha+1}^{(M)}: \\ \mathcal{L}(h_W) - \mathcal{L}(f^\circ) \leq \epsilon^2}} \beta\lambda\|W\|_{\mathcal{H}_{\alpha+1}^{(M)}}^2 \lesssim \beta(\epsilon^*)^2.$$

Finally, since $\tilde{\nu}_\beta^{(M)}$ is a marginal distribution of $\tilde{\nu}_\beta^{(\infty)}$, it holds that

$$-\log \tilde{\nu}_\beta^{(M)}(\{W \in \mathcal{H}^{(M)} : \|W\|_{\mathcal{H}^{(M)}} \leq \epsilon\}) \leq -\log \tilde{\nu}_\beta^{(\infty)}(\{W \in \mathcal{H} : \|W\|_{\mathcal{H}} \leq \epsilon\}).$$

Therefore, as long as $M \geq (\epsilon^*)^{-1/[\theta(\alpha+1)]}$, the rate of $\epsilon^*$ is not deteriorated from $M = \infty$. In other words, if $M \geq \min\{\lambda^{\tilde{\alpha}/2[\theta(\alpha+1)]}\beta^{1/2[\theta(\alpha+1)]}, \lambda^{-\theta/2[\theta(\alpha+1)]}, n^{1/2[\theta(\alpha+1)]}\}$, the bound (13) holds. $\qquad\square$

**Remark 2.** *Suzuki (2020) showed Proposition 4 under a condition $\alpha > 1/2$. However, this is used only to ensure Assumption 3. In our setting, we can show Assumption 3 holds directly and thus we may omit the condition $\alpha > 1/2$.*

### B.2 PROOFS OF PROPOSITION 1, THEOREM 2 AND COROLLARY 1

Here, we give the proofs of Proposition 1 and Theorem 2 simultaneously.

*Proof of Proposition 1 and Theorem 2.* Let $\bar{R} = (2\sum_{m=1}^\infty a_m R + U)^2$. Then, we can easily check that $(y_i - f_W(x_i))^2 \leq \bar{R}$. As stated above, we use Propositions 3 and 4 to show the statements.

First, we show Proposition 1 for the dynamics of $W_k^{(M)}$ for any $1 \leq M \leq \infty$. However, it suffices to show the statement only for $M = \infty$ because the finite dimensional version can be seen as a

specific case of the infinite dimensional one. Actually, the dynamics of $W_k^{(M)}$ is same as that of $\iota(\tilde{W}_k)$ where $\tilde{W}_k \in \mathcal{H}$ obeys the following dynamics:

$$\tilde{W}_{k+1} = S_\eta \left( \tilde{W}_k - \eta \nabla \widehat{\mathcal{L}}(f_{\iota(\tilde{W}_k)}) + \sqrt{\frac{2\eta}{\beta}} \xi_k \right).$$

This is because $f_{\iota(\tilde{W}_k)}$ is determined by only the first $M$ components $\iota(\tilde{W}_k)$, $\iota(\nabla \widehat{\mathcal{L}}(f_{\iota(\tilde{W}_k)})) = \nabla_{W^{(M)}} \widehat{\mathcal{L}}(f_{W^{(M)}})|_{W^{(M)} = \iota(\tilde{W}_k)}$ and $S_\eta$ is a diagonal operator. Since the components of $\tilde{W}_k$ with indexes higher than $M$ does not affect the objective, smoothness of the objective is not lost. The stationary distribution $\pi_\infty^{(M)}$ of the continuous dynamics corresponding to $W^{(M)}$ is a probability measure on $\mathcal{H}^{(M)}$ that satisfies

$$\frac{\mathrm{d}\pi_\infty^{(M)}}{\mathrm{d}\nu_\beta^{(M)}}(W^{(M)}) \propto \exp(-\beta \widehat{\mathcal{L}}(f_{W^{(M)}})),$$

where $\nu_\beta^{(M)}$ is the Gaussian measure on $\mathbb{R}^{M \times (d+2)}$ with mean 0 and covariance $(\beta A^{(M)})^{-1}$. We can notice that this is the marginal distribution of the stationary distribution of the continuous time counterpart of $\tilde{W}_k$: $\mathrm{d}\tilde{\pi}_\infty(\tilde{W}) \propto \exp(-\beta \widehat{\mathcal{L}}(f_{\iota(\tilde{W})}))\mathrm{d}\nu_\beta$. Therefore, we just need to consider an infinite dimensional one. For this reasoning, we show the convergence for the original infinite dimensional dynamics $(W_k)_{k=1}^\infty$. The convergence of the finite dimensional one $(W_k^{(M)})_{k=1}^\infty$ can be shown by the same manner using the argument above.

To show Proposition 1, we use Propositions 3. To do so, we need to check validity of Assumptions 3. First, we check Assumption 3. Assumption 3-(i) is ensured by Assumption 1. Next, we check Assumption 3-(ii). The boundedness of the gradient can be shown as follows:

$$\|\nabla \widehat{\mathcal{L}}(f_W)\|_\mathcal{H}^2$$
$$= \sum_{m=1}^\infty \left( \left\| \frac{1}{n} \sum_{i=1}^n 2(f_W(x_i) - y_i)\bar{w}_{2,m} a_m [x_i; 1]\sigma_m'(w_{1,m}^\top [x_i; 1]) \right\|^2 \right.$$
$$\left. + \left| \frac{1}{n} \sum_{i=1}^n 2(f_W(x_i) - y_i)a_m \tanh'(w_{2,m}/R)\sigma_m(w_{1,m}^\top [x_i; 1]))_{m=1}^\infty \right|^2 \right)$$
$$\leq \sum_{m=1}^\infty 4\bar{R}R^2 a_m^2 (d+1)C_\sigma^2 + 4\bar{R}a_m^2$$
$$(\because |f_W(x_i) - y_i| \leq \bar{R}, \ \|\sigma_m'\|_\infty \leq C_\sigma, \ \|\tanh'\|_\infty \leq 1)$$
$$\leq 4\bar{R}[R^2 C_\sigma^2 (d+1) + 1] \sum_{m=1}^\infty a_m^2 < \infty.$$

Similarly, we can show the Lipschitz continuity of the gradient as

$$\|\nabla \widehat{\mathcal{L}}(f_W) - \nabla \widehat{\mathcal{L}}(f_{W'})\|_\mathcal{H}^2$$
$$\leq \sum_{m=1}^\infty \mu_m^{-2\alpha_1} \mu_m^{2\alpha_1} \left\{ 4\bar{R}a_m^2 (d+1)C_\sigma^2[(w_{2,m} - w_{2,m}')^2 + R^2 \|w_{1,m} - w_{1,m}'\|^2] \right.$$
$$\left. + 4\bar{R}a_m^2[(w_{2,m} - w_{2,m}')^2/R^2 + C_\sigma^2 (d+1)\|w_{1,m} - w_{1,m}'\|^2] \right\} \quad (\because \|\tanh''\|_\infty \leq 1)$$
$$\leq 4\bar{R}[(d+1)C_\sigma^2(1 + R^2) + 1/R^2 + C_\sigma^2(d+1)] \max_{m \in \mathbb{N}}\{\mu_m^{-2\alpha_1} a_m^2\}$$
$$\times \sum_{m=1}^\infty \mu_m^{2\alpha_1}[(w_{2,m} - w_{2,m}')^2 + \|w_{1,m} - w_{1,m}'\|^2]$$
$$\lesssim \|W - W'\|_{\mathcal{H}_{-\alpha_1}}^2.$$

We can also verify Assumption 3-(iii) in a similar way. Then, we have verified Assumption 3. Therefore, we may apply Proposition 3, and then we obtain Proposition 1.

Next, we show Theorem 2 by using Proposition 4. For that purpose, we need to we verify Assumption 4. The first condition can be verified as

$$E_{X,Y}[((Y - f_W(X))^2 - (Y - f^o(X))^2)^2]$$
$$= E_{X,\epsilon}[((f^o(X) + \epsilon - f_W(X))^2 - \epsilon^2)^2]$$
$$= E_X[((f^o(X) - f_W(X))^2 + 2\epsilon(f^o(X) - f_W(X)))^2]$$
$$= E_X[(f^o(X) - f_W(X))^4 + 2\epsilon(f^o(X) - f_W(X))(f^o(X) - f_W(X))^2 + \epsilon^2(f^o(X) - f_W(X))^2]$$
$$= \|f^o - f_W\|_\infty^2 E_X[(f^o(X) - f_W(X))^2] + U^2 E_X[(f^o(X) - f_W(X))^2]$$
$$\le \bar{R} E_X[(f^o(X) - f_W(X))^2] = \bar{R}(\mathcal{L}(f_W) - \mathcal{L}(f^o)).$$

The second condition can be checked as follows. Note that

$$E_{Y|X=x}\left(\exp\left\{-\frac{\beta}{n}[(Y - f_W(x))^2 - (Y - f^o(x))^2]\right\}\right)$$
$$= E_\epsilon\left(\exp\left[-\frac{\beta}{n}(f^o(x) - f_W(x))^2 - 2\epsilon(f_W(x) - f^o(x))]\right\}\right)$$
$$= \exp\left[-\frac{\beta}{n}(f^o(x) - f_W(x))^2\right] E_\epsilon\left\{\exp\left[\frac{2\beta}{n}\epsilon(f_W(x) - f^o(x))\right]\right\}$$
$$\le \exp\left[-\frac{\beta}{n}(f^o(x) - f_W(x))^2\right] \exp\left[\frac{1}{8}\frac{4\beta^2}{n^2}4U^2(f_W(x) - f^o(x))^2\right].$$

Thus, under the condition $\beta \le n/(2U^2)$, the right hand side can be upper bounded by

$$\exp\left[-\frac{\beta}{n}\left(1 - 2\frac{U^2\beta}{n}\right)(f_W(x) - f^o(x))^2\right] \le 1.$$

Next, we check the third and fourth conditions. Noting that

$$\nabla_W h_W(X)$$
$$= \left(a_m\overline{(\mu_m^{-\alpha/2} w_{2,m})}\mu_m^{-\alpha/2}[x_i; 1]\sigma'_m(\mu_m^{-\alpha/2} w_{1,m}^\top[x_i; 1]),\right.$$
$$\left. a_m\mu_m^{-\alpha/2}\tanh'(\mu_m^{-\alpha/2} w_{2,m}/R)\sigma_m(\mu_m^{-\alpha/2} w_{1,m}^\top[x_i; 1]))_{m=1}^\infty\right)_{m=1}^\infty,$$

we have that

$$\|\nabla_W h_W(X)\|_{\mathcal{H}}^2$$
$$\le \sum_{m=1}^\infty a_m^2\mu_m^{-\alpha}[(d+1)R^2C_\sigma^2 + 1]$$
$$\le [(d+1)R^2C_\sigma^2 + 1]\sum_{m=1}^\infty \mu_m^{-\alpha+2\alpha_1}$$
$$\le [(d+1)R^2C_\sigma^2 + 1]c_\mu^{-\alpha+2\alpha_1}\sum_{m=1}^\infty m^{-2(-\alpha+2\alpha_1)} =: C_1 < \infty$$
$$(\because -\alpha + 2\alpha_1 = \alpha_1 > 1/2),$$

and

$$\|\nabla_W h_W(X) - \nabla_W h_{W'}(X)\|_{\mathcal{H}}^2$$
$$\le \sum_{m=1}^\infty a_m^2\mu_m^{-\alpha}(d+1)[\mu_m^{-\alpha}(w_{2,m} - w'_{2,m})^2 + R^2\mu_m^{-\alpha}\|w_{1,m} - w'_{1,m}\|^2]$$
$$+ a_m^2\mu_m^{-\alpha}[\mu_m^{-\alpha}(w_{2,m} - w'_{2,m})^2/R^2 + C_\sigma^2(d+1)\mu_m^{-\alpha}\|w_{1,m} - w'_{1,m}\|^2]$$
$$\le \sum_{m=1}^\infty a_m^2\mu_m^{-2\alpha}[(d+1)(1+R^2) + 1/R^2 + C_\sigma^2(d+1)][\|w_{1,m} - w'_{1,m}\|^2 + (w_{2,m} - w'_{2,m})^2]$$

$$\leq c_\mu^{2\alpha_1} \max_m\{\mu_m^{2(\alpha_1-\alpha)}\}[(d+1)(1+R^2)+1/R^2+C_\sigma^2(d+1)]\|W-W'\|_{\mathcal{H}}^2 =: C_2\|W-W'\|_{\mathcal{H}}^2,$$

for a constant $0 < C_2 < \infty$. Therefore, it holds that

$$|h_W(X)-h_{W'}(X)|^2 \leq C_1\|W-W'\|_{\mathcal{H}}^2,$$

which yields the forth condition, and we also have

$$\|\nabla_W\ell(Y,h_W(X))-\nabla_W\ell(Y,h_{W'}(X))\|_{\mathcal{H}}^2$$
$$=\|2(h_W(X)-Y)\nabla_W h_W(X)-2(h_{W'}(X)-Y)\nabla_W h_{W'}(X)\|_{\mathcal{H}}^2$$
$$\leq 2\|2(h_W(X)-Y)(\nabla_W h_W(X)-\nabla_W h_W(X))\|_{\mathcal{H}}^2$$
$$+2\|2(h_W(X)-h_{W'}(X))\nabla_W h_{W'}(X)\|_{\mathcal{H}}^2$$
$$\leq 8\bar{R}C_2\|W-W'\|_{\mathcal{H}}^2+8C_1^2\|W-W'\|_{\mathcal{H}}^2 \lesssim \|W-W'\|_{\mathcal{H}}^2,$$

which yields the third condition.

Since $f^\circ \in \mathcal{F}_\gamma$, there exists $W^* \in \mathcal{H}_\gamma$ such that $f^\circ = f_{W^*}$. Therefore, applying Proposition 4 with $\alpha = \alpha_1$ ($\tilde{\alpha} = 1/[2(\alpha_1+1)]$) and $\theta = \gamma/(1+\alpha_1)$ (since $\gamma < 1/2+\alpha_1$, the condition $\theta < 1-\tilde{\alpha}$ is satisfied), we obtain that for $M \geq \min\left\{\lambda^{1/4\gamma(\alpha_1+1)}\beta^{1/2\gamma}, \lambda^{-1/2(\alpha_1+1)}, n^{1/2\gamma}\right\}$, the following excess risk bound holds:

$$\mathrm{E}_{D^n}\left[\mathrm{E}_{W_k^{(M)}}[\mathcal{L}(W_k^{(M)})|D_n]-\mathcal{L}(f^*)\right] \lesssim \max\left\{(\lambda\beta)^{\frac{2\tilde{\alpha}/\theta}{1+\tilde{\alpha}/\theta}}n^{-\frac{1}{1+\tilde{\alpha}/\theta}}, \lambda^{-\tilde{\alpha}}\beta^{-1}, \lambda^\theta, 1/n\right\}+\Xi_k.$$

Finally, by noting $\mathcal{L}(W_k^{(M)})-\mathcal{L}(f^*) = \|f_{W_k^{(M)}}-f^*\|_{L_2(P_X)}^2$, we obtain the assertion. $\square$

Finally, we give the proof of Corollary 1.

*Proof of Corollary 1.* Note that

$$f_W(x)$$
$$=\sum_{m=1}^\infty a_m\bar{w}_{2,m}\sigma_m(w_{1,m}^\top[x;1])$$
$$=\sum_{m=1}^\infty \mu_m^{\alpha_1}\bar{w}_{2,m}\mu_m^{q\alpha_2}\mu_m^{-q\alpha_2}\mu_m^{s\alpha_2}\sigma(\mu_m^{-\alpha_2}w_{1,m}^\top[x;1]) \quad (\because a_m = \mu_m^{\alpha_1}, \ b_m = \mu_m^{\alpha_2})$$
$$=\sum_{m=1}^\infty \mu_m^{\alpha_1+q\alpha_2}\bar{w}_{2,m}\mu_m^{-(s-q)\alpha_2}\sigma(\mu_m^{-\alpha_2}w_{1,m}^\top[x;1]).$$

Therefore, we may redefine $\alpha_1' \leftarrow \alpha_1+q\alpha_2$ and $s' \leftarrow s-q$ so that we obtain another representation of the model $\mathcal{F}_\gamma$:

$$\mathcal{F}_\gamma = \left\{f_W(x)=\sum_{m=1}^\infty \mu_m^{\alpha_1'}\bar{w}_{2,m}\check{\sigma}_m(w_{1,m}^\top[x;1]) \ \middle| \ W \in \mathcal{H}_\gamma, \|W\|_{\mathcal{H}_\gamma} \leq 1\right\},$$

where $\check{\sigma}_m(\cdot) = \mu_m^{-s'\alpha_2}\sigma(\mu_m^{-\alpha_2}\cdot)$. Note that the condition $0 \leq q \leq s-3$ gives $s-q \geq 3$. Therefore, Assumptions 3 and 4 are valid even for the redefined parameters $\alpha_1'$, $s'$ and $\check{\sigma}_m$ instead of $\alpha_1$, $s$ and $\sigma_m$. Therefore, we can apply Theorem 2 by simply replacing $\alpha_1$ by $\alpha_1' = \alpha_1+q\alpha_2$. $\square$

