# OpenReview forum: "Benefit of deep learning with non-convex noisy gradient descent: Provable excess risk bound and superiority to kernel methods"
_ICLR.cc/2021/Conference — ICLR 2021 Spotlight_

### Official Review · AnonReviewer1 · 2020-10-21
**Interesting and to the point results, technically very demanding, so difficult to verify correctness.**

**Rating:** 8
**Confidence:** 3

**Review:**

=========================

Summary:

The paper shows that a two-layer neural network (although an extension to deeper models seem unproblematic) may outperform a class of linear functions in terms of the excess risk learning rate, and in a minimax optimality analysis, and when approximating a target function from the neural network class. The paper essentially shows that linear functions have a problem with the non-convexity of the neural network class, and approximate the slow rate of 1/(n)^(1/2) for increasing dimension. A neural network trained with noisy stochastic gradient descent on the other hand has a faster rate, depending on several parameters.

=========================

=========================

Pros:

- Well written and polished paper.

- Technically sounds as far as I can tell. (Randomly checked some parts in more detail.)

- Setting and results may be interesting for a large audience.

- Main results and message of the paper are to the point.

=========================

=========================

Cons:

Very technical and on some parts I would have liked some more intuition and discussion. See detailed feedback and also questions for rebuttal.

=========================

=========================

Scoring:

Overall I think this is a worthwhile contribution in understanding the difference in deep and shallow learning, and as the paper is very sound I will vote for accept. I will acknowledge, however, that there is a flurry of related work, as it is a very popular topic, and I can not vouch for the novelty of this contribution. The authors, however, covered much ground in that regard.

=========================

=========================

Questions for rebuttal:

It appears to me that the neural networks are not part of the linear functions class, and thus having a neural network target makes the linear functions being misspecified. Is that true? If so, does that play a role in the learning rate gap? In case it is not true, what is the essential difference then between the linear functions and the neural networks? Regarding that, what is phi_i in the definition of linear models?

Instead of noisy gradient descent you actually use semi-implicit euler scheme for optimization, do you have any thoughts on how that might effect actual performance?

As far as I can see your current analysis does not hold for relu-activations, how easy might an extension to that be?

Are you aware of any lower bounds for the neural network case, are your rates optimal?

=========================

=========================

Additional feedback:

The result that the minimiax rate of linear functions over a space F is the same as over its convex hull was not known to me. For me it would have been very useful if you could provide some intuition on why that is the case.

You show that the rate of the neural network is independent of the dimension. Do you have any intuition on why that is the case?

Under Equation (5), instead of "more faster,...,more faster" write "the faster,..., the faster"

---

> ### Author Response · Authors · 2020-11-17
> **Reply to Reviewer #1 (1)**
>
> Thank you very much for your positive feedback and insightful comments.
>
> Q. It appears to me that the neural networks are not part of the linear functions class, and thus having a neural network target makes the linear functions being misspecified. Is that true? If so, does that play a role in the learning rate gap?
> A.
> This interpretation is a little bit different from what is actually going on. Indeed, we can construct a model in which the neural network model can be included, for example, we can construct a kernel function for which the corresponding RKHS includes the neural network model. For example, if we set $k(x,x') = \int \sum_{m=1}^\infty c_m \sigma_m(w_{1,m}^\top [x;1])\sigma_m(w_{1,m}^\top [x';1]) d \nu_m(w_{1,m}),$ where $c_m$ is a positive constant with $\sum_m c_m = 1$ and $\nu_m(\cdot)$ is a probability measure. Then, the neural network model is included in this RKHS (or at least, every element in the neural network model can be approximated by an element in the RKHS with any precision). Another example is a Sobolev space. Since the neural network model is smooth, all functions in the neural network model is included in a Sobolev space with an appropriate smoothness parameter. Therefore, it is easy to find a linear space that includes the neural network model, and thus the comparison with linear estimators including a kernel ridge regression is not completely unfair. However, as our theorem states, such an RKHS with a kernel function including ${\mathcal{F}}{\gamma}$ becomes unnecessarily large because it must cover any $\sigma_m(w_{1,m}^\top [x;1])$ with different $w_{1,m}$, then the convergence rate becomes slower. On the other hand, the neural network can appropriately pick up only one $w_{1,m}$ for each $m$, which makes the estimator much more efficient than kernel methods. This intuition can be mathematically formulated as the convex hull argument that we have used in the main text.
>
> Q. what is $\varphi_i$?
> A.
> First, we would like to emphasize that the linear estimator admits "any" measurable (and L^2-integrable) function as $\varphi_i$. You may choose anything you like. Of course, you can choose the "best" function as $\varphi_i$ that would minimize the excess risk. One important example is a kernel ridge regression. Since the kernel ridge regression can be written as $\hat{f}(x) = \sum_{i=1}^n y_i ((K_X + \lambda I)^{-1}\mathbf{k}(x))_i$, then we can set $\varphi_i(x_1,\dots,x_n,x) = ((K_X + \lambda I)^{-1}\mathbf{k}(x))_i$ (we can check that the right hand side is a function of $(x_1,...,x_n,x)$). It is also possible to use such a kernel function that was introduced above.
>
> Q. Instead of noisy gradient descent you actually use semi-implicit euler scheme for optimization, do you have any thoughts on how that might effect actual performance?
> A.
> We think its actual impact is quite marginal. In practice, we use a finite dimensional approximation ($W^{(M)}$) where the width $M$ is not so large (indeed M is less than the sample size $n$). In such a regime, the difference between those two schemes is small if we choose the step size $eta$ sufficiently small. Therefore, we think that the usual Euler scheme instead of the semi-implicit Euler scheme would work well in practice.
>
> Q. As far as I can see your current analysis does not hold for relu-activations, how easy might an extension to that be?
> A.
> You are absolutely true. We think we might be able to extend the analysis to non-differentially activation functions such as ReLU because adding noise to the dynamics is equivalent to smoothing the objective function as shown by [R1]. This would be far from trivial, but we think it is possible.
>
> [R1] Bobby Kleinberg, Yuanzhi Li, Yang Yuan. "An Alternative View: When Does SGD Escape Local Minima?", Proceedings of the 35th International Conference on Machine Learning, PMLR 80:2698-2707, 2018.
>
>
> Q. Are you aware of any lower bounds for the neural network case, are your rates optimal?
> A.
> Thank you very much for bringing up an important issue. Unfortunately, the minimax optimal rate for the class $\mathcal{F}_\gamma$ is not known. More precisely, we have a rough lower bound $n^{-\frac{\gamma + \alpha_1 + s\alpha_2 + 1/4}{\gamma + \alpha_1 + s\alpha_2 + 1/2}}$ so far, but we are not completely sure whether this is tight. We would like to defer deriving the minimax optimal rate as a future work.

---

> > ### Comment · AnonReviewer1 · 2020-11-24
> > **Response**
> >
> > Dear authors,
> >
> > thank you for you clarifications, the few concerns that I had were answered. Additionally I think the addition of the intuitions will help readers to appreciate the results more. Thus I will remain with my vote for accepting.
> >
> > Best Regards

---

> ### Author Response · Authors · 2020-11-17
> **Reply to Reviewer #1 (2)**
>
> (This is a continuation of "Reply to Reviewer 4 (1)". We are sorry for the long reply.)
>
> Additional feedback:
> Q. It would have been very useful if you could provide some intuition on why that is the case.
> A.
> Intuitively, since the linear estimator is linear to the observations $(y_i)_{i=1}^n$ of outputs, a simple application of Jensen's inequality yields that its worst case error on the convex hull of the function class $\mathcal{F}$ does not increase compared with that on the original one $\mathcal{F}$. Please look at Hayakawa & Suzuki (2020) for its rigorous proof. We have added this sentence in the revised version.
>
> Q. You show that the rate of the neural network is independent of the dimension. Do you have any intuition on why that is the case?
> A. This is because, for each m, we only need to specify one parameter $(w_{1,m},w_{2,m})$ that have (d+2)-dimensional, and we do not need to estimate a linear combination of each neuron. However, a linear estimator should cover such a linear combination by their model.
>
> Q. Under Equation (5), instead of "more faster,...,more faster" write "the faster,..., the faster"
> A. Thank you very much for pointing out this typo. We have fixed this in the revised version.

---

### Official Review · AnonReviewer2 · 2020-10-26
**Comments to "Benefit of deep learning with non-convex noisy gradient descent: ..."**

**Rating:** 8
**Confidence:** 4

**Review:**

#### General comments
This paper aims at proving superiority of neural network models to any linear estimators, including kernel methods.  To attain this purpose,  this paper focuses on  two layer neural network class with an infinite width. For the non-parametric regression models within this neural network class, this paper establishes a sharp excess risk error of the least square methods with  noisy gradient descent update, although such optimization may be heavily non-convex. Moreover, a lower bound of  all linear estimators under the $L_2$-norm   are accordingly given when the true function is within the two layer neural network class, thereby showing superiority to kernel  methods.  Overall，the contribution of this paper is obvious and the literature review is full to some extent.
This paper is organized well and stated clearly.

#### Specific Comments
（1）After Theorem 1, the sentence "for relative high dimensional settings, this lower bound becomes close to a slow rate $\Omega(1/\sqrt{n})$, which corresponds to the curse of dimensionality. "  I argue  that this sentence may be uncorrected,
since the mentioned rate is independent of the input dimension, which is not a real curse of dimensionality.
(2)  A constraint on  $f_{W}$ should be added, otherwise, it is impossible to identify $a_m$ and $\bar{w}_{2,m}$ simultaneously.
(3) What is the role of noisy term in NSGD algorithm, is it a similar conclusion when the standard SGD is applied?
(4）What  is the additional difficulty encountered when analyzing a thin but deep neural network?

---

> ### Author Response · Authors · 2020-11-17
> **Reply to Reviewer #2**
>
> Thank you very much for your positive feedback and suggestive comments.
> We reply to your specific comments one by one as follows.
>
> (1) Please look at the derived lower bound again. It is $n^{-\frac{2\tilde{\beta} + d}{2\tilde{\beta} + 2d}}$ that includes the input dimension $d$ in the rate. Actually, if we increase $d$ to infinity, the exponent $\frac{2\tilde{\beta} + d}{2\tilde{\beta} + 2d}$ converges to $1/2$ yielding the convergence rate $1/\sqrt{n}$.
> (2) Please note that $a_m$ is a fixed constant which we do not need to estimate. Moreover, what we need to estimate is {\it not} the parameters itself but the function $f_W$. Therefore, identifiability of parameters is not required in our setting. Actually, in our proof, we are not showing the convergence of estimated parameter to a "true" parameter but we have shown only the convergence of estimated "function" to the true function $f_{W^*}$.
> (3) Thank you for bringing up an important point. The noisy term is required to get out of a local optimal. Without the noisy term, the optimization dynamics can stack in a local minimum. Another role is that it makes the dynamics of the solution behaves as if it is generated from a Bayes posterior distribution, which enables us to analyze the convergence rate of the excess risk.
> (4) We think it is more or less straight forward to extend the upper bound of the excess risk of neural network to a thin deep neural network. On the other hand, it would be more involved to derive a tight lower bound of the excess risk of linear estimators.

---

### Official Review · AnonReviewer3 · 2020-10-28
**Official Blind Review #3**

**Rating:** 6
**Confidence:** 2

**Review:**

Summary:

The paper aims to demonstrate the superiority of deep learning methods against kernel methods by comparing their excess risk bounds. In particular, the authors first derive the minimax lower bound for linear estimators by assuming that the target function can be represented by a teacher neural network, which implies that the linear estimators suffer from curse of dimensionality. Then the authors further derive a dimension-independent upper bound of the noisy gradient descent method for overparametrized two-layer neural networks, which theoretically confirms the benefit of deep learning methods in terms of convergence rates. The paper is well written and also interesting to read. Overall, I vote for accepting.


Concerns:
1. In the teacher-student setting, what is the minimax rate for any estimator of $f^0$ instead of just linear estimators? Or is the upper bound for the noisy gradient descent method minimax optimal?
2. Traditionally we impose smoothness assumption on the target function directly (e.g. Holder space). So what is the main advantage of this teacher-student setting?
3. In Theorem 1, I feel a little bit confused why the dimension $d$ also appears in the numerator, which is different from classical lower bounds. For example, If we assume that $f^0$ belongs to a Sobolev space of order $r$, then the minimax rate of excess risk will be $n^{-\frac{2r}{2r+d}}$, which goes to 0 as $d$ goes to infinity. Do I have any misunderstanding?

---

> ### Author Response · Authors · 2020-11-17
> **Reply to Reviewer #3**
>
> Thank you very much for your insightful comments.
>
> Q. In the teacher-student setting, what is the minimax rate for any estimator of  instead of just linear estimators? Or is the upper bound for the noisy gradient descent method minimax optimal?
> A.
> Thank you very much for bringing up an important issue. Unfortunately, the minimax optimal rate for the class $\mathcal{F}_\gamma$ is not known. More precisely, we have a rough lower bound $n^{-\frac{\gamma + \alpha_1 + s\alpha_2 + 1/4}{\gamma + \alpha_1 + s\alpha_2 + 1/2}}$, but we are not completely sure whether this is tight. We would like to defer deriving the minimax optimal rate as a future work.
>
> Q. Traditionally we impose smoothness assumption on the target function directly (e.g. Holder space). So what is the main advantage of this teacher-student setting?
> A.
> Yes, the Holder/Sobolev space is a typical setting to characterize the convergence rate via the smoothness. However, the geometry of these typical spaces are not non-convex and there does not hardly appear difference between deep and shallow. On the other hand, using the teacher-student setting induces more sparsity which plays an important role to obtain a non-convex geometry of the model. This is related to the feature extraction structure of neural network. Via the feature extraction ability, the neural network focuses more on a specific part of the input that induces non-convexity. On the other hand, the Holder space directly uses the whole information of the input and its smoothness is uniform over all input $x$, which results in a convex model.
>
> Q. In Theorem 1, I feel a little bit confused why the dimension d also appears in the numerator, which is different from classical lower bounds. For example, If we assume that f^o belongs to a Sobolev space of order s, then the minimax rate of excess risk will be $n^{-2s/(2s+d)}$, which goes to 0 as  goes to infinity. Do I have any misunderstanding?
> A.
> This is indeed a good point. The convergence rate on the Sobolev space indicates that the complexity of the space is much more sensitive to the input dimension $d$ than the neural network model. Actually, the deep learning approach is not affected by the curse of dimensionality. The phenomenon that $d$ appears in the numerator of the minimax rate can be found in the lower bound of linear estimators in the Besov space too. For a Besov space $B^\beta_{p,q}$ with $\beta > d(1/p - 1/2)$ with $p < 2$, the lower bound of linear estimators is $n^{-\frac{2(\beta - d(1/p - 1/2))}{2(\beta - d(1/p - 1/2)) + d}} = n^{-\frac{2 s + d}{2 s + 2d}}$ where we set $s = \beta - d/p$. In this case, the dimension $d$ appears in the denominator. By carefully looking at the proof, we can notice that the role of $s$ and $\tilde{\beta}$ are the same. Therefore, we think our lower bound is natural.

---

### Official Review · AnonReviewer4 · 2020-10-28
**Interesting work but requires more refined explanations and discussions**

**Rating:** 7
**Confidence:** 4

**Review:**

This paper is theoretical sound and well organized. This paper shows that the Bayes estimator with Gaussian prior can outperform the linear estimators (including kernel regression and k-NN), which I believe is indeed interesting and important.

Besides, I would like to raise the following comments and questions.

1. The network function is different from the commonly-used one. For example, the authors need to clip the output weights using tanh function. The authors state that the reason is to ensure the boundness condition of the network function. What if using a standard parameterization of a two-layer network but performing projected (noisy) gradient descent?

2. It seems that the goal of all estimators is to recover the teacher networks. However, the Bayes estimator actually uses the same network structure as the teacher one. It is more interesting to investigate the case where the underline teacher network is independent of the learned network (e.g., using an overparameterized network to learn a smaller network).

3. In the description of hat f, the authors may need to clearly state the definition of the function \phi_i.

4. Does the result in Theorem 2 hold for any f^{\circ}?

5. In Proposition, the convergence results look similar to the following paper, while in their paper, the right-hand side of (6) converges to O(\eta^{1/2}) when k\eta goes to infinity. Could you briefly discuss why in this paper, this quantity is in the order of O(\eta^{1/2-a})?

Xu, Pan, et al. "Global Convergence of Langevin Dynamics Based Algorithms for Nonconvex Optimization." arXiv preprint arXiv:1707.06618 (2017).

6. Besides, I feel that the comparison with Raginsky et al., 2017 and Erdogdu et al., 2018 after Proposition may not be fair. In particular, the convergence results in these papers are derived based on different assumptions, thus their dependencies on the dimension are not directly comparable.

7. Moreover, it may not be appropriate to state that “NGD achieves a fast convergence rate”, it seems that the spectral gap \Lambda^* still has an exponential dependency on the parameter beta (shown in Proposition 3), which will be set as beta = \Theta(n) in Theorem 2. This implies that the noisy gradient descent may require exponential time to output a good solution.

8. In Theorem 2, can we view the expected error between F_{W_k^{(M)}}  and f^{\circ} as a variant of generalization error (in expectation)? If this is the case, can we somehow apply the results in the following paper, and obtain a O(n^{-1}) generalization error bound for the Langevin dynamic gradient algorithm (if considering finite dimension case)?

Mou, Wenlong, et al. "Generalization bounds of SGLD for non-convex learning: Two theoretical viewpoints." Conference on Learning Theory. 2018.

9. A suggestion: It is better to present Section B.1 before the first part of Section B, since the proof of Proposition 1, Theorem 2, and Corollary 1 largely rely on the assumptions and propositions in Section B.1.

10. Lastly, the authors may also want to include the following two NTK papers in the introduction section.

Zou, Difan, et al. "Gradient descent optimizes over-parameterized deep ReLU networks." Machine Learning 109.3 (2020): 467-492.

Cao, Yuan, and Quanquan Gu. "Generalization bounds of stochastic gradient descent for wide and deep neural networks." Advances in Neural Information Processing Systems. 2019.

---

> ### Author Response · Authors · 2020-11-17
> **Reply to Reviewer 4 (1)**
>
> Thank you very much for your insightful comments.
>
> Q. The network function is different from the commonly-used one. What if using a standard parameterization of a two-layer network but performing projected (noisy) gradient descent?
> A.
> The projected gradient descent is a proximal gradient descent with an indicator function, but the current theory of the infinite dimensional gradient Langevin dynamics does not cover a non-differentiable and unbounded objective. Thus, it is difficult to rigorously show its convergence in that setting. However, by the analogy from the finite dimensional analysis, it should not affect so much to the result in practice.
> A more critical assumption in our analysis is that the scaling factors $a_m$ and $b_m$ converge to 0 as $m \to \infty$. This makes the model include a function with high frequency components that leads separation between deep learning and linear estimators.
>
> Q. It seems that the goal of all estimators is to recover the teacher networks. However, the Bayes estimator actually uses the same network structure as the teacher one. It is more interesting to investigate the case where the underline teacher network is independent of the learned network.
> A.
> We agree with your opinion that a setting where a student model is different from the teacher model would be more interesting. On the other hand, the teacher-student model is quite basic and commonly used in the literature. In that sense, we consider that it is natural to discuss optimality of estimators on the teacher-student model. We may find a better neural network based estimator that outperforms the estimator we considered in the paper, but it does not contradict a fact that any linear estimator suffers from a curse of dimensionality but an appropriate deep learning approach does not. We think showing this fact in a simplest setting (such as teacher-student model) would be quite important in the literature.
>
> Q. In the description of hat f, the authors may need to clearly state the definition of the function \phi_i.
> A.
> We would like to emphasize that the definition of a linear estimator admits {\it any} $\varphi_i$. The only assumption for $\varphi_i$ is that it is measurable with respect to $x$ and $x_1,\dots,x_n$. Our lower bound of the excess risk of linear estimators is applicable uniformly to all estimator that has the form of $\hat{f}$ (of course, that includes the kernel ridge regression). Therefore, you may consider the "optimal" choice of $\varphi_i$ that would "minimize" the excess risk in some sense. Since we do not need to restrict the shape of $\varphi_i$, we think that the result of Theorem 1 is a strong statement. We have added a note that explicitly tells $\varphi_i$ can be any measurable function right after the definition of the linear estimator.
>
> Q. Does the result in Theorem 2 hold for any f^{\circ}?
> A. Yes, the result holds for any $f^\circ \in \mathcal{F}_\gamma$ uniformly.
>
> Q. In Proposition, the convergence results look similar to the following paper, while in their paper, the right-hand side of (6) converges to $O(\eta^{1/2})$ when $k\eta$ goes to infinity.
> A.
> We guess you intended $O(\eta)$ in their paper. If so, this is because of infinite dimensional setting and the regularization term. In our setting the regularization term is $\|W\|^2_{\mathcal{H}1} = \sum_m (w_{1,m}^2 + w_{2,m}^2)/k^2$, but if we employ $\|W\|^2_{\mathcal{H}{p/2}} = \sum_m (w_{1,m}^2 + w_{2,m}^2)/(k^p)$ for $p > 1$ then that term should be $\eta^{(p-1)/p}$. On the other hand, the finite dimensional setting corresponds to taking the limit of $p \to \infty$ which leads to $\eta^{(p-1)/p} \to \eta$ that recovers the finite dimensional result ($O(\eta)$) as shown by Xu et al. (2017).

---

> ### Author Response · Authors · 2020-11-17
> **Reply to Reviewer 4 (2)**
>
> (This is a continuation of "Reply to Reviewer 4 (1)". We are sorry for the long reply.)
>
> Q. Besides, I feel that the comparison with Raginsky et al., 2017 and Erdogdu et al., 2018 after Proposition may not be fair.
> A. As we have remarked above, the convergence of the infinite dimensional version is guaranteed due to the existence of regularization term. Roughly speaking, the regularization term makes the analysis similar to a finite dimensional one. However, it is far from trivial. Actually, the dependency of the step size \eta is changed from $O(\eta)$ to $O(\eta^{1/2-a})$ in exchange for enabling the infinite dimensional analysis. In that sense, the assumption is different but we think it is quite important to see the connection between finite dimensional analysis and infinite dimensional one.
>
> Q. Moreover, it may not be appropriate to state that gNGD achieves a fast convergence rate, it seems that the spectral gap $\Lambda^*$ still has an exponential dependency on the parameter beta (shown in Proposition 3).
> A.
> We guess the terminology "fast convergence rate" was a bit confusing. We used this terminology to indicate the convergence rate of the "statistical convergence rate" of the excess risk with respect to the sample size $n$. We did not intended to indicate the algorithmic convergence rate of NGD with respect to the number of iterations $k$. As you pointed out the spectral gap $\Lambda^*_\eta$ depends on the parameter $\beta$ exponentially. This is stated in the definition of the spectral gap, but since this paper's focus is more on the statistical analysis, we did not mention it explicitly due to the space limitation. On the other hand, we realized that to avoid the confusion you pointed out, we had better to add one sentence which explicitly shows this point. Accordingly, we have added such a sentence in the revised version right after Proposition 1.
>
> Q. In Theorem 2, can we view the expected error between F_{W_k^{(M)}} and f^{\circ} as a variant of generalization error (in expectation)? If this is the case, can we somehow apply the results in the following paper, and obtain a O(n^{-1}) generalization error bound for the Langevin dynamic gradient algorithm (if considering finite dimension case)?
> A. Thank you for pointing out an important issue. We cannot directly obtain our excess risk bound from the generalization error bound ($L(\hat{f}) - \hat{L}(\hat{f})$). As you noticed, the excess risk can be bounded by using the relation $L(\hat{f}) - L(f^\circ) = (L(\hat{f}) - \hat{L}(\hat{f})) + (\hat{L}(\hat{f}) - \hat{L}(f^\circ)) + (\hat{L}(f^\circ) - L(f^\circ))$ and the fact that the second term of the right hand side can be small for $\hat{f}$ with small training error and the third term can be bounded by the Hoeffding's inequality. However, this approach *does not* yield our excess risk bound. Instead, we have used a variant of the *local* Rademacher complexity technique (we need several modification to deal with the Bayes estimator). This analysis utilizes the strong convexity and smoothness of the squared loss that enables us to utilize a fact that an estimator $\hat{f}$ with small excess risk is close to the true function $f^\circ$. By using this, we can obtain a faster learning rate and, consequently, we can compare the convergence rate.
>
> Q. A suggestion: It is better to present Section B.1 before the first part of Section B, since the proof of Proposition 1, Theorem 2, and Corollary 1 largely rely on the assumptions and propositions in Section B.1.
> A. Thank you for your careful reading and insightful suggestion. We have changes the order of the sections following your suggestion.
>
> Q. Lastly, the authors may also want to include the following two NTK papers in the introduction section.
> A. Thank you for your suggestion. We could not cite several important references due to the space limitation. We have included the references that you have suggested in the revised version.

---

### Author Response · Authors · 2020-11-17
**To all reviewers**

Thank you very much for your careful reading and suggestive comments.
We have revised our paper according to your comments. The major changes are as follows:
1. We have added a remark to the definition of the linear estimator that states that $\varphi_i$ can be any measurable function.
2. A intuition about why the minimax risk of the linear estimators is same as that on the convex hull of the target function class is added.
3. We have added a notion about the convergence rate of the algorithm, especially about the spectral gap.
4. We have added some missing important references.

Sincerely yours,
Authors.

---

### Decision · Program_Chairs · 2021-01-07
**Final Decision**

**Decision:**

Accept (Spotlight)

**Comment:**

This paper analyzes deep networks optimized using non-convex noisy gradient descent. The main result shows that in a teacher-student setting, the excess risk converges in a fast-rate and is stronger than any linear estimators (which include kernel methods). The paper also gives a convergence rate result that depends on some spectral gaps (which can be very small) but not on dimension. Overall the paper is interesting. It should probably emphasize that the dependency on spectral gaps (and the fact that they could be exponentially small) on the convergence as the current abstract suggests efficient convergence.